# Cal-QL: Calibrated Offline RL Pre-Training for Efficient Online Fine-Tuning

**Mitsuhiko Nakamoto**[1][*]   **Yuexiang Zhai**[1][*]   **Anikait Singh**[1]   **Max Sobol Mark**[2]

**Yi Ma**[1]   **Chelsea Finn**[2]   **Aviral Kumar**[1]   **Sergey Levine**[1]

[1]UC Berkeley   [2]Stanford University

## Abstract

A compelling use case of offline reinforcement learning (RL) is to obtain a policy initialization from existing datasets followed by fast online fine-tuning with limited interaction. However, existing offline RL methods tend to behave poorly during fine-tuning. In this paper, we study the fine-tuning problem in the context of conservative offline RL methods and we devise an approach for learning an effective initialization from offline data that also enables fast online fine-tuning capabilities. Our approach, calibrated Q-learning (Cal-QL), accomplishes this by learning a conservative value function initialization that underestimates the value of the learned policy from offline data, while also ensuring that the learned Q-values are at a reasonable scale. We refer to this property as calibration, and define it formally as providing a lower bound on the true value function of the learned policy and an upper bound on the value of some other (suboptimal) reference policy, which may simply be the behavior policy. We show that a conservative offline RL algorithm that also learns a calibrated value function leads to effective online fine-tuning, enabling us to take the benefits of offline initializations in online fine-tuning. In practice, Cal-QL can be implemented on top of the conservative Q learning (CQL) [32] for offline RL within a one-line code change. Empirically, Cal-QL outperforms state-of-the-art methods on **9/11** fine-tuning benchmark tasks that we study in this paper. Code and video are available at https://nakamotoo.github.io/Cal-QL

## 1   Introduction

Modern machine learning successes follow a common recipe: pre-training models on general-purpose, Internet-scale data, followed by fine-tuning the pre-trained initialization on a limited amount of data for the task of interest [22, 7]. How can we translate such a recipe to sequential decision-making problems? A natural way to instantiate this paradigm is to utilize offline reinforcement learning (RL) [37] for initializing value functions and policies from static datasets, followed by online fine-tuning to improve this initialization with limited active interaction. If successful, such a recipe might enable efficient online RL with much fewer samples than current methods that learn from scratch.

Many algorithms for offline RL have been applied to online fine-tuning. Empirical results across such works suggest a counter-intuitive trend: policy initializations obtained from more effective offline RL methods tend to exhibit worse online fine-tuning performance, even within the same task (see Table 2 of Kostrikov et al. [31] & Figure 4 of Xiao et al. [57]). On the other end, online RL methods training from scratch (or RL from demonstrations [53], where the replay buffer is seeded with the offline data) seem to improve online at a significantly faster rate. However,

---

[*]Equal contributions. Corresponding authors: Mitsuhiko Nakamoto, Yuexiang Zhai, and Aviral Kumar ({nakamoto, simonzhai}@berkeley.edu, aviralku@andrew.cmu.edu)

37th Conference on Neural Information Processing Systems (NeurIPS 2023).

these online methods require actively collecting data by rolling out policies from scratch, which inherits similar limitations to naïve online RL methods in problems where data collection is expensive or dangerous. Overall, these results suggest that it is challenging to devise an offline RL algorithm that both acquires a good initialization from prior data and also enables efficient fine-tuning.

How can we devise a method to learn an effective policy initialization that also improves during fine-tuning? Prior work [32, 6] shows that one can learn a good offline initialization by optimizing the policy against a *conservative* value function obtained from an offline dataset. But, as we show in Section 4.1, conservatism alone is insufficient for efficient online fine-tuning. Conservative methods often tend to "unlearn" the policy initialization learned from offline data and waste samples collected via online interaction in recovering this initialization. We find that the "unlearning" phenomenon is a consequence of the fact that value estimates produced via conservative methods can be significantly lower than the ground-truth return of *any* valid policy. Having Q-value estimates that do not lie on a similar scale as the return of a valid policy is problematic. Because once fine-tuning begins, actions executed in the environment for exploration that are actually worse than the policy learned from offline data could erroneously

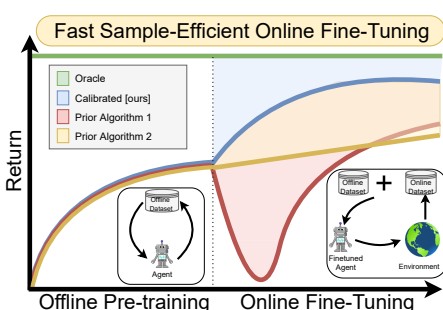

Figure 1: We study **offline RL pre-training followed by online RL fine-tuning**. Some prior offline RL methods tend to exhibit slow performance improvement in this setting (yellow), resulting in worse asymptotic performance. Others suffer from initial performance degradation once online fine-tuning begins (red), resulting in a high cumulative regret. We develop an approach that "*calibrates*" the learned value function to attain a fast improvement with a smaller regret (blue).

appear better, if their ground-truth return value is larger than the learned conservative value estimate. Hence, subsequent policy optimization will degrade the policy performance until the method recovers.

If we can ensure that the conservative value estimates learned using the offline data are *calibrated*, meaning that these estimates are on a similar scale as the true return values, then we can avoid the unlearning phenomenon caused by conservative methods (see the formal definition in 4.1). Of course, we cannot enforce such a condition perfectly, since it would require eliminating all errors in the value function. Instead, we devise a method for ensuring that the learned values upper bound the true values of some *reference policy* whose values can be estimated more easily (e.g., the behavior policy), while still lower bounding the values of the learned policy. Though this does not perfectly ensure that the learned values are correct, we show that it still leads to sample-efficient online fine-tuning. Thus, our practical method, **calibrated Q-learning (Cal-QL)**, learns conservative value functions that are "calibrated" against the behavior policy, via a simple modification to existing conservative methods.

The main contribution of this paper is Cal-QL, a method for acquiring an offline initialization that facilitates online fine-tuning. Cal-QL aims to learn conservative value functions that are calibrated with respect to a reference policy (e.g., the behavior policy). Our analysis of Cal-QL shows that Cal-QL attains stronger guarantees on cumulative regret during fine-tuning. In practice, Cal-QL can be implemented on top of conservative Q-learning [32], a prior offline RL method, without any additional hyperparameters. We evaluate Cal-QL across a range of benchmark tasks from [10], [51] and [44], including robotic manipulation and navigation. We show that Cal-QL matches or outperforms the best methods on all tasks, in some cases by 30-40%.

## 2 Related Work

Several prior works suggest that online RL methods typically require a large number of samples [50, 54, 61, 26, 64, 18, 38] to learn from scratch. We can utilize offline data to accelerate online RL algorithms. Prior works do this in a variety of ways: incorporating the offline data into the replay buffer of online RL [48, 53, 23, 52], utilizing auxiliary behavioral cloning losses with policy gradients [46, 27, 67, 66], or extracting a high-level skill space for downstream online RL [17, 1]. While these methods improve the sample efficiency of online RL from scratch, as we will also show in our results, they do not eliminate the need to actively roll out poor policies for data collection.

To address this issue, a different line of work first runs offline RL for learning a good policy and value initialization from the offline data, followed by online fine-tuning [45, 30, 41, 3, 56, 36, 42]. These

approaches typically employ offline RL methods based on policy constraints or pessimism [12, 49, 16, 15, 30, 51, 36] on the offline data, then continue training with the same method on a combination of offline and online data once fine-tuning begins [43, 28, 62, 32, 4]. Although pessimism is crucial for offline RL [25, 6], using pessimism or constraints for fine-tuning [45, 30, 41] slows down fine-tuning or leads to initial unlearning, as we will show in Section 4.1. In effect, these prior methods either fail to improve as fast as online RL or lose the initialization from offline RL. We aim to address this limitation by understanding some conditions on the offline initialization that enable fast fine-tuning.

Our work is most related to methods that utilize a pessimistic RL algorithm for offline training but incorporate exploration in fine-tuning [36, 42, 56]. In contrast to these works, our method aims to learn a better offline initialization that enables standard online fine-tuning. Our approach fine-tunes naïvely without ensembles [36] or exploration [42] and, as we show in our experiments, this alone is enough to outperform approaches that employ explicit optimism during data collection.

## 3 Preliminaries and Background

The goal in RL is to learn the optimal policy for an MDP $\mathcal{M} = (\mathcal{S}, \mathcal{A}, P, r, \rho, \gamma)$. $\mathcal{S}, \mathcal{A}$ denote the state and action spaces. $P(s'|s, a)$ and $r(s, a)$ are the dynamics and reward functions. $\rho(s)$ denotes the initial state distribution. $\gamma \in (0, 1)$ denotes the discount factor. Formally, the goal is to learn a policy $\pi : \mathcal{S} \mapsto \mathcal{A}$ that maximizes cumulative discounted value function, denoted by $V^\pi(s) = \frac{1}{1-\gamma} \sum_t \mathbb{E}_{a_t \sim \pi(s_t)} [\gamma^t r(s_t, a_t) | s_0 = s]$. The Q-function of a given policy $\pi$ is defined as $Q^\pi(s, a) = \frac{1}{1-\gamma} \sum_t \mathbb{E}_{a_t \sim \pi(s_t)} [\gamma^t r(s_t, a_t) | s_0 = s, a_0 = a]$, and we use $Q_\theta^\pi$ to denote the estimate of the Q-function of a policy $\pi$ as obtained via a neural network with parameters $\theta$.

Given access to an offline dataset $\mathcal{D} = \{(s, a, r, s')\}$ collected using a behavior policy $\pi_\beta$, we aim to first train a good policy and value function using the offline dataset $\mathcal{D}$ alone, followed by an online phase that utilizes online interaction in $\mathcal{M}$. Our goal during fine-tuning is to obtain the optimal policy with the smallest number of online samples. This can be expressed as minimizing the **cumulative regret** over rounds of online interaction: $\text{Reg}(K) := \mathbb{E}_{s \sim \rho} \sum_{k=1}^K \left[ V^\star(s) - V^{\pi^k}(s) \right]$. As we demonstrate in Section 7, existing methods face challenges in this setting.

Our approach will build on the conservative Q-learning (CQL) [32] algorithm. CQL imposes an additional regularizer that penalizes the learned Q-function on out-of-distribution (OOD) actions while compensating for this pessimism on actions seen within the training dataset. Assuming that the value function is represented by a function, $Q_\theta$, the training objective of CQL is given by

$$\min_\theta \alpha \underbrace{\left( \mathbb{E}_{s \sim \mathcal{D}, a \sim \pi} [Q_\theta(s, a)] - \mathbb{E}_{s, a \sim \mathcal{D}} [Q_\theta(s, a)] \right)}_{\text{Conservative regularizer } \mathcal{R}(\theta)} + \frac{1}{2} \mathbb{E}_{s, a, s' \sim \mathcal{D}} \left[ \left( Q_\theta(s, a) - \mathcal{B}^\pi \bar{Q}(s, a) \right)^2 \right],$$

(3.1)

where $\mathcal{B}^\pi \bar{Q}(s, a)$ is the backup operator applied to a delayed target Q-network, $\bar{Q}$: $\mathcal{B}^\pi \bar{Q}(s, a) := r(s, a) + \gamma \mathbb{E}_{a' \sim \pi(a'|s')} [\bar{Q}(s', a')]$. The second term is the standard TD error [40, 13, 20]. The first term $\mathcal{R}(\theta)$ (in blue) is a conservative regularizer that aims to prevent overestimation in the Q-values for OOD actions by minimizing the Q-values under the policy $\pi(a|s)$, and counterbalances by maximizing the Q-values of the actions in the dataset following the behavior policy $\pi_\beta$.

## 4 When Can Offline RL Initializations Enable Fast Online Fine-Tuning?

A starting point for offline pre-training and online fine-tuning is to simply initialize the value function with one that is produced by an existing offline RL method and then perform fine-tuning. However, we empirically find that initializations learned by many offline RL algorithms can perform poorly during fine-tuning. We will study the reasons for this poor performance for the subset of conservative methods to motivate and develop our approach for online fine-tuning, calibrated Q-learning.

### 4.1 Empirical Analysis

Offline RL followed by online fine-tuning typically poses non-trivial challenges for a variety of methods. While analysis in prior work [45] notes challenges for a subset of offline RL methods, in Figure 2, we evaluate the fine-tuning performance of a variety of prior offline RL methods (CQL [32], IQL [30], TD3+BC [11], AWAC [45]) on a particular diagnostic instance of a visual pick-and-place task with a distractor object and sparse binary rewards [51], and find that all methods struggle to attain the best possible performance, quickly. More details about this task are in Appendix B.

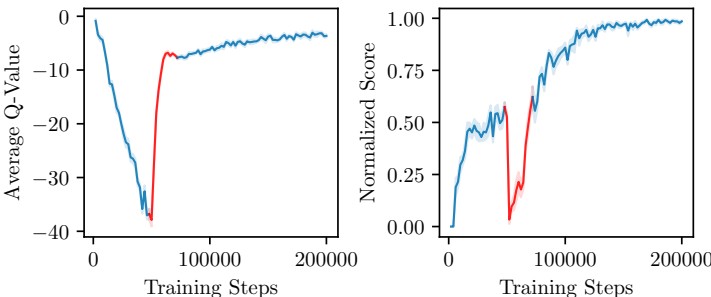

Figure 3: **The evolution of the average Q-value and the success rate of CQL over the course of offline pre-training and online fine-tuning.** Fine-tuning begins at 50K steps. The red-colored part denotes the period of performance recovery which also coincides with the period of Q-value adjustment.

While the offline Q-function initialization obtained from all methods attains a similar (normalized) return of around 0.5, they suffer from difficulties during fine-tuning: TD3+BC, IQL, AWAC attain slow asymptotic performance and CQL unlearns the offline initialization, followed by spending a large amount of online interaction to recover the offline performance again, before any further improvement. This initial unlearning appears in multiple tasks as we show in Appendix F. In this work, we focus on developing effective fine-tuning strategies on top of conservative methods like CQL. To do so, we next aim to understand the potential reason behind the initial unlearning in CQL.

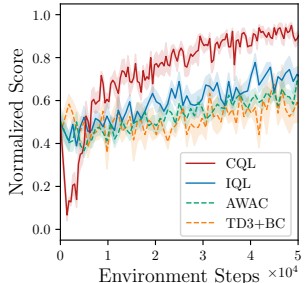

Figure 2: **Multiple prior offline RL algorithms suffer from difficulties** during fine-tuning including poor asymptotic performance and initial unlearning.

**Why does CQL unlearn initially?** To understand why CQL unlearns initially, we inspect the learned Q-values averaged over the dataset in Figure 3. Observe that the Q-values learned by CQL in the offline phase are *much* smaller than their ground-truth value (as expected), but these Q-values drastically jump and adjust in scale when fine-tuning begins. In fact, we observe that performance recovery (red segment in Figure 3) *coincides* with a period where the range of Q-values changes to match the true range. This is as expected: as a conservative Q-function experiences new online data, actions much worse than the offline policy on the rollout states appear to attain higher rewards compared to the highly underestimated offline Q-function, which in turn deceives the policy optimizer into unlearning the initial policy. We illustrate this idea visually in Figure 4. Once the Q-function has adjusted and the range of Q-values closely matches the true range, then fine-tuning can proceed normally, after the dip.

**To summarize,** our empirical analysis indicates that methods existing fine-tuning methods suffer from difficulties such as initial unlearning or poor asymptotic performance. In particular, we observed that conservative methods can attain good asymptotic performance, but "waste" samples to correct the learned Q-function. Thus, in this paper, we attempt to develop a good fine-tuning method that builds on top of an existing conservative offline RL method, CQL, but aims to "calibrate" the Q-function so that the initial dip in performance can be avoided.

### 4.2 Conditions on the Offline Initialization that Enable Fast Fine-Tuning

Our observations from the preceding discussion motivate two conclusions in regard to the offline Q-initialization for fast fine-tuning: **(a)** methods that learn **conservative** Q-functions can attain good asymptotic performance, and **(b)** if the learned Q-values closely match the range of ground-truth Q-values on the task, then online fine-tuning does not need to devote samples to unlearn and then recover the offline initialization. One approach to formalize this intuition of Q-values lying on a similar scale as the ground-truth Q-function is via the requirement that the conservative Q-values learned by the conservative offline RL method must be lower-bounded by the ground-truth Q-value of a sub-optimal reference policy. This will prevent conservatism from learning overly small Q-values. We will refer to this property as "calibration" with respect to the reference policy.

**Definition 4.1** (Calibration). *An estimated Q-function $Q_\theta^\pi$ for a given policy $\pi$ is said to be calibrated with respect to a reference policy $\mu$ if $\mathbb{E}_{a\sim\pi}\left[Q_\theta^\pi(s,a)\right] \geq \mathbb{E}_{a\sim\mu}\left[Q^\mu(s,a)\right] := V^\mu(s), \forall s \in D$.*

If the learned Q-function $Q_\theta^\pi$ is calibrated with respect to a policy $\mu$ that is worse than $\pi$, it would prevent unlearning during fine-tuning that we observed in the case of CQL. This is because the

policy optimizer would not unlearn $\pi$ in favor of a policy that is worse than the reference policy $\mu$ upon observing new online data as the expected value of $\pi$ is constrained to be larger than $V^\mu$: $\mathbb{E}_{a \sim \pi} [Q_\theta^\pi(s, a)] \geq V^\mu(s)$. Our practical approach Cal-QL will enforce calibration with respect to a policy $\mu$ whose ground-truth value, $V^\mu(s)$, can be estimated reliably without bootstrapping error (e.g., the behavior policy induced by the dataset). This is the key idea behind our method (as we will discuss next) and is visually illustrated in Figure 4.

## 5 Cal-QL: Calibrated Q-Learning

Our approach, calibrated Q-learning (Cal-QL) aims to learn a conservative and calibrated value function initializations from an offline dataset. To this end, Cal-QL builds on CQL [32] and then constrains the learned Q-function to produce Q-values larger than the Q-value of a reference policy $\mu$ per Definition 4.1. In principle, our approach can utilize many different choices of reference policies, but for developing a practical method, we simply utilize the behavior policy as our reference policy.

**Calibrating CQL.** We can constrain the learned Q-function $Q_\theta^\pi$ to be larger than $V^\mu$ via a simple change to the CQL training objective shown in Equation 3.1: masking out the push down of the learned Q-value on out-of-distribution (OOD) actions in CQL if the Q-function is not calibrated, i.e., if $\mathbb{E}_{a \sim \pi} [Q_\theta^\pi(s, a)] \leq V^\mu(s)$. Cal-QL modifies the CQL regularizer, $\mathcal{R}(\theta)$ in this manner:

$$\mathbb{E}_{s \sim \mathcal{D}, a \sim \pi} \left[ \max \left( Q_\theta(s, a), V^\mu(s) \right) \right] - \mathbb{E}_{s, a \sim \mathcal{D}} \left[ Q_\theta(s, a) \right], \tag{5.1}$$

where the changes from standard CQL are depicted in red. As long as $\alpha$ (in Equation 3.1) is large, for any state-action pair where the learned Q-value is smaller than $Q^\mu$, the Q-function in Equation 5.1 will upper bound $Q^\mu$ in a tabular setting. Of course, as with any practical RL method, with function approximators and gradient-based optimizers, we cannot guarantee that we can enforce this condition for every state-action pair, but in our experiments, we find that Equation 5.1 is sufficient to enforce the calibration in expectation over the states in the dataset.

**Pseudo-code and implementation details.** Our implementation of Cal-QL directly builds on the implementation of CQL from Geng [14]. We present a pseudo-code for Cal-QL in Appendix A. Additionally, we list the hyperparameters $\alpha$ for the CQL algorithm and our baselines for each suite of tasks in Appendix C. Following the protocol in prior work [30, 52], the practical implementation of Cal-QL trains on a mixture of the offline data and the new online data, weighted in some proportion during fine-tuning. To get $V^\mu(s)$, we can fit a function approximator $Q_\theta^\mu$ or $V_\theta^\mu$ to the return-to-go values via regression, but we observed that also simply utilizing the return-to-go estimates for tasks that end in a terminal was sufficient for our use case. We show in Section 7, how this simple *one-line* change to the objective drastically improves over prior fine-tuning results.

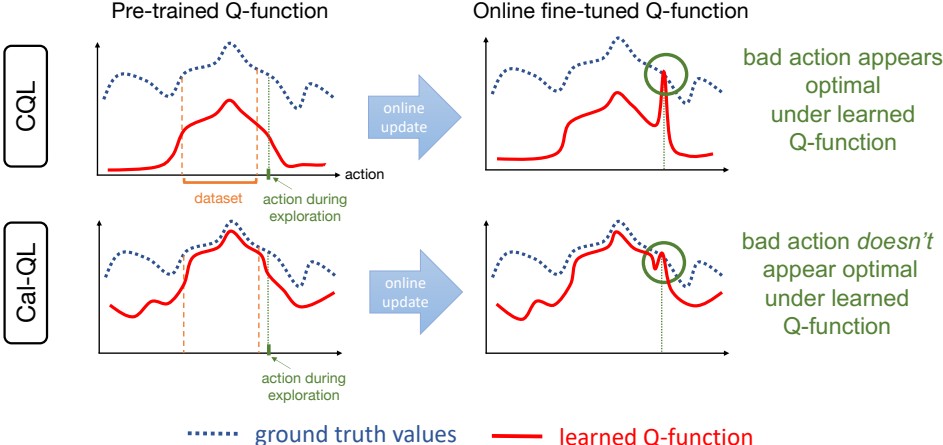

Figure 4: **Intuition behind policy unlearning with CQL and the idea behind Cal-QL.** The plot visualizes a slice of the learned Q-function and the ground-truth values for a given state. Erroneous peaks on suboptimal actions (x-axis) arise when updating CQL Q-functions with online data. This in turn can lead the policy to deviate away from high-reward actions covered by the dataset in favor of erroneous new actions, resulting in deterioration of the pre-trained policy. In contrast, Cal-QL corrects the scale of the learned Q-values by using a reference value function, such that actions with worse Q-values than the reference value function do not erroneously appear optimal in fine-tuning.

# 6 Theoretical Analysis of Cal-QL

We will now analyze the cumulative regret attained over online fine-tuning, when the value function is pre-trained with Cal-QL, and show that enforcing calibration (Defintion 4.1) leads to a favorable regret bound during the online phase. Our analysis utilizes tools from Song et al. [52], but studies the impact of calibration on fine-tuning. We also remark that we simplify the treatment of certain aspects (e.g., how to incorporate pessimism) as it allows us to cleanly demonstrate the benefits of calibration.

**Notation & terminology.** In our analysis, we will consider an idealized version of Cal-QL for simplicity. Specifically, following prior work [52] under the bilinear model [9], we will operate in a finite-horizon setting with a horizon $H$. We denote the learned Q-function at each learning iteration $k$ for a given $(s, a)$ pair and time-step $h$ by $Q_\theta^k(s, a)$. For any given policy $\pi$, let $C_\pi \geq 1$ denote the concentrability coefficient such that $C_\pi := \max_{f \in \mathcal{C}} \frac{\sum_{h=0}^{H-1} \mathbb{E}_{s,a \sim d_h^\pi}[\mathcal{T}f_{h+1}(s,a) - f_h(s,a)]}{\sqrt{\sum_{h=0}^{H-1} \mathbb{E}_{s,a \sim \nu_h}(\mathcal{T}f_{h+1}(s,a) - f_h(s,a))^2}}$, i.e., a coefficient that quantifies the distribution shift between the policy $\pi$ and the dataset $\mathcal{D}$, in terms of the ratio of Bellman errors averaged under $\pi$ and the dataset $\mathcal{D}$. Note that $\mathcal{C}$ represents the Q-function class and we assume $\mathcal{C}$ has a bellman-bilinear rank [9] of $d$. We also use $C_\pi^\mu$ to denote the concentrability coefficient over a subset of *calibrated* Q-functions w.r.t. a reference policy $\mu$: $C_\pi^\mu := \max_{f \in \mathcal{C}, f(s,a) \geq Q^\mu(s,a)} \frac{\sum_{h=0}^{H-1} \mathbb{E}_{s,a \sim d_h^\pi}[\mathcal{T}f_{h+1}(s,a) - f_h(s,a)]}{\sqrt{\sum_{h=0}^{H-1} \mathbb{E}_{s,a \sim \nu_h}(\mathcal{T}f_{h+1}(s,a) - f_h(s,a))^2}}$, which provides $C_\pi^\mu \leq C_\pi$. Similar to $\mathcal{C}$, let $d_\mu$ denote the bellman bilinear rank of $\mathcal{C}_\mu$ – the calibrated Q-function class w.r.t. the reference policy $\mu$. Intuitively, we have $\mathcal{C}_\mu \subset \mathcal{C}$, which implies that $d_\mu \leq d$. The formal definitions are provided in Appendix H.2. We will use $\pi^k$ to denote the arg-max policy induced by $Q_\theta^k$.

**Intuition.** We intuitively discuss how calibration and conservatism enable Cal-QL to attain a smaller regret compared to not imposing calibration. Our goal is to bound the cumulative regret of online fine-tuning, $\sum_k \mathbb{E}_{s_0 \sim \rho}[V^{\pi^\star}(s_0) - V^{\pi^k}(s_0)]$. We can decompose this expression into two terms:

$$\text{Reg}(K) = \underbrace{\sum_{k=1}^{K} \mathbb{E}_{s_0 \sim \rho}\left[V^\star(s_0) - \max_a Q_\theta^k(s_0, a)\right]}_{(i) := \text{ miscalibration}} + \underbrace{\sum_{k=1}^{K} \mathbb{E}_{s_0 \sim \rho}\left[\max_a Q_\theta^k(s_0, a) - V^{\pi^k}(s_0)\right]}_{(ii) := \text{ overestimation}}. \quad (6.1)$$

This decomposition of regret into terms (i) and (ii) is instructive. Term (ii) corresponds to the amount of over-estimation in the learned value function, which is expected to be small if a conservative RL algorithm is used for training. Term (i) is the difference between the ground-truth value of the optimal policy and the learned Q-function and is negative if the learned Q-function were calibrated against the optimal policy (per Definition 4.1). Of course, this is not always possible because we do not know $V^\star$ a priori. But note that when Cal-QL utilizes a reference policy $\mu$ with a high value $V^\mu$, close to $V^\star$, then the learned Q-function $Q_\theta$ is calibrated with respect to $Q^\mu$ per Condition 4.1 and term (i) can still be controlled. Therefore, controlling this regret requires striking a balance between learning a calibrated (term (i)) and conservative (term (ii)) Q-function. We now formalize this intuition and defer the detailed proof to Appendix H.6.

**Theorem 6.1** (Informal regret bound of Cal-QL). *With high probability, Cal-QL obtains the following bound on total regret accumulated during online fine-tuning:*

$$\text{Reg}(K) = \widetilde{O}\left(\min\left\{C_{\pi^\star}^\mu H \sqrt{dK \log(|\mathcal{F}|)}, \ K\mathbb{E}_\rho[V^\star(s_0) - V^\mu(s_0)] + H\sqrt{d_\mu K \log(|\mathcal{F}|)}\right\}\right),$$

*where $\mathcal{F}$ is the functional class of the Q-function.*

**Comparison to Song et al. [52].** Song et al. [52] analyzes an online RL algorithm that utilizes offline data without imposing conservatism or calibration. We now compare Theorem 6.1 to Theorem 1 of Song et al. [52] to understand the impact of these conditions on the final regret guarantee. Theorem 1 of Song et al. [52] presents a regret bound: $\text{Reg}(K) = \widetilde{O}\left(C_{\pi^\star} H \sqrt{dK \log(|\mathcal{F}|)}\right)$ and we note some improvements in our guarantee, that we also verify via experiments in Section 7.3: **(a)** for the setting where the reference policy $\mu$ contains near-optimal behavior, i.e., $V^\star - V^\mu \lesssim O(H\sqrt{d \log(|\mathcal{F}|)/K})$, Cal-QL can enable a tighter regret guarantee compared to Song et al. [52]; **(b)** as we show in Appendix H.3, the concentrability coefficient $C_{\pi^\star}^\mu$ appearing in our guarantee is no larger than the one that appears in Theorem 1 of Song et al. [52], providing another source of improvement; and **(c)** finally, in the case where the reference policy has broad coverage *and* is highly sub-optimal, Cal-QL reverts back to the guarantee from [52], meaning that Cal-QL improves upon this prior work.

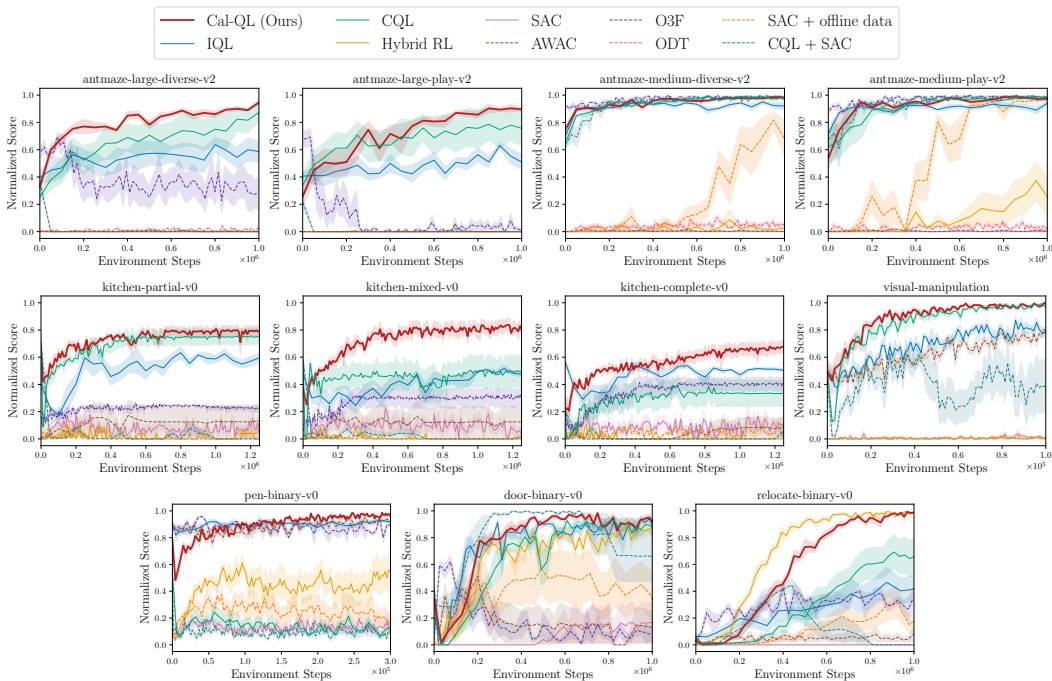

Figure 6: **Online fine-tuning after offline initialization on the benchmark tasks**. The plots show the online fine-tuning phase *after* pre-training for each method (except SAC-based approaches which are not pre-trained). Observe that Cal-QL consistently matches or exceeds the speed and final performance of the best prior method and is the only algorithm to do so across all tasks. (6 seeds)

## 7  Experimental Evaluation

The goal of our experimental evaluation is to study how well Cal-QL can facilitate sample-efficient online fine-tuning. To this end, we compare Cal-QL with several other state-of-the-art fine-tuning methods on a variety of offline RL benchmark tasks from D4RL [10], Singh et al. [51], and Nair et al. [45], evaluating performance before and after fine-tuning. We also study the effectiveness of Cal-QL on higher-dimensional tasks, where the policy and value function must process raw image observations. Finally, we perform empirical studies to understand the efficacy of Cal-QL with different dataset compositions and the impact of errors in the reference value function estimation.

**Offline RL tasks and datasets.** We evaluate Cal-QL on a number of benchmark tasks and datasets used by prior works [30, 45] to evaluate fine-tuning performance: **(1)** The `AntMaze` tasks from D4RL [10] that require controlling an ant quadruped robot to navigate from a starting point to a desired goal location in a maze. The reward is +1 if the agent reaches within a pre-specified small radius around the goal and

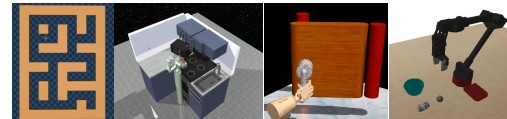

Figure 5: **Tasks:** We evaluate Cal-QL on a diverse set of benchmark problems: `AntMaze` and `Frankakitchen` domains from [10], `Adroit` tasks from [45] and a vision-based robotic manipulation task from [34].

0 otherwise. **(2)** The `FrankaKitchen` tasks from D4RL require controlling a 9-DoF Franka robot to attain a desired configuration of a kitchen. To succeed, a policy must complete four sub-tasks in the kitchen within a single rollout, and it receives a binary reward of +1/0 for every sub-task it completes. **(3)** Three `Adroit` dexterous manipulation tasks [47, 30, 45] that require learning complex manipulation skills on a 28-DoF five-fingered hand to **(a)** manipulate a pen in-hand to a desired configuration (`pen-binary`), **(b)** open a door by unlatching the handle (`door-binary`), and **(c)** relocating a ball to a desired location (`relocate-binary`). The agent obtains a sparse binary +1/0 reward if it succeeds in solving the task. Each of these tasks only provides a narrow offline dataset consisting of 25 demonstrations collected via human teleoperation and additional trajectories collected by a BC policy. Finally, to evaluate the efficacy of Cal-QL on a task where we learn from raw visual observations, we study **(4)** a pick-and-place task from prior work [51, 34] that requires learning to pick a ball and place it in a bowl, in the presence of distractors. Additionally, we compared Cal-QL on D4RL locomotion tasks (`halfcheetah`, `hopper`, `walker`) in Appendix D.

**Comparisons, prior methods, and evaluation protocol.** We compare Cal-QL to running online SAC [21] from scratch, as well as prior approaches that leverage offline data. This includes naïvely fine-tuning offline RL methods such as CQL [32] and IQL [30], as well as fine-tuning with AWAC [45], O3F [42] and online decision transformer (ODT) [65], methods specifically designed for offline RL followed by online fine-tuning. In addition, we also compare to a baseline that trains SAC [21] using both online data and offline data (denoted by "SAC + offline data") that mimics DDPGfD [53] but utilizes SAC instead of DDPG. We also compare to Hybrid RL [52], a recently proposed method that improves the sample efficiency of the "SAC + offline data" approach, and "CQL+SAC", which first pre-train with CQL and then run fine-tuning with SAC on a mixture of offline and online data without conservatism. More details of each method can be found in Appendix C. We present learning curves for online fine-tuning and also quantitatively evaluate each method on its ability to improve the initialization learned from offline data measured in terms of **(i)** final performance after a pre-defined number of steps per domain and **(ii)** the cumulative regret over the course of online fine-tuning. In Section 7.2, we run Cal-QL with a higher update-to-data (UTD) ratio and compare it to RLPD [2], a more sample-efficient version of "SAC + offline data".

## 7.1  Empirical Results

We first present a comparison of Cal-QL in terms of the normalized performance before and after fine-tuning in Table 1 and the cumulative regret in a fixed number of online steps in Table 2. Following the protocol of [10], we normalize the average return values for each domain with respect to the highest possible return (+4 in FrankaKitchen; +1 in other tasks; see Appendix C.1 for more details).

**Cal-QL improves the offline initialization significantly.** Observe in Table 1 and Figure 6 that while the performance of offline initialization acquired by Cal-QL is comparable to that of other methods such as CQL and IQL, Cal-QL is able to improve over its offline initialization the most by **106.9%** in aggregate and achieve the best fine-tuned performance in **9 out of 11** tasks.

**Cal-QL enables fast fine-tuning.** Observe in Table 2 that Cal-QL achieves the smallest regret on **8 out of 11** tasks, attaining an average regret of 0.22 which improves over the next best method (IQL) by **42%**. Intuitively, this means that Cal-QL does not require running highly sub-optimal policies. In tasks such as `relocate-binary`, Cal-QL enjoys the fast online learning benefits associated with naïve online RL methods that incorporate the offline data in the replay buffer (SAC + offline data and Cal-QL are the only two methods to attain a score of $\geq 90\%$ on this task) unlike prior offline RL methods. As shown in Figure 6, in the `kitchen` and `antmaze` domains, Cal-QL brings the benefits of fast online learning together with a good offline initialization, improving drastically on the regret metric. Finally, observe that the initial unlearning at the beginning of fine-tuning with conservative methods observed in Section 4.1 is greatly alleviated in all tasks (see Appendix F for details).

| Task | CQL | IQL | AWAC | O3F | ODT | CQL+SAC | Hybrid SRL | SAC+od | SAC | Cal-QL (Ours) |
|---|---|---|---|---|---|---|---|---|---|---|
| large-diverse | 25 → 87 | 40 → 59 | 00 → 00 | 59 → 28 | 00 → 01 | 36 → 00 | → 00 | → 00 | → 00 | 33 → **95** |
| large-play | 34 → 76 | 41 → 51 | 00 → 00 | 68 → 01 | 00 → 00 | 21 → 00 | → 00 | → 00 | → 00 | 26 → **90** |
| medium-diverse | 65 → **98** | 70 → 92 | 00 → 00 | 92 → 97 | 00 → 03 | 64 → **98** | → 02 | → 68 | → 00 | 75 → **98** |
| medium-play | 62 → 98 | 72 → 94 | 00 → 00 | 89 → **99** | 00 → 05 | 67 → 98 | → 25 | → 96 | → 00 | 54 → 97 |
| partial | 71 → 75 | 40 → 60 | 01 → 13 | 11 → 22 | - | 71 → 00 | → 00 | → 07 | → 03 | 67 → **79** |
| mixed | 56 → 50 | 48 → 48 | 02 → 12 | 06 → 33 | - | 59 → 01 | → 01 | → 00 | → 02 | 38 → **80** |
| complete | 13 → 34 | 57 → 50 | 01 → 08 | 17 → 41 | - | 21 → 06 | → 00 | → 05 | → 06 | 22 → **68** |
| pen | 55 → 13 | 88 → 92 | 88 → 92 | 91 → 89 | - | 48 → 10 | → 54 | → 17 | → 11 | 79 → **99** |
| door | 22 → 88 | 41 → 88 | 29 → 13 | 04 → 08 | - | 29 → 66 | → 88 | → 39 | → 17 | 35 → **92** |
| relocate | 06 → 69 | 06 → 45 | 06 → 08 | 03 → 35 | - | 01 → 00 | → **99** | → 16 | → 00 | 03 → 98 |
| manipulation | 50 → 97 | 49 → 81 | 50 → 73 | - | - | 42 → 41 | → 00 | → 01 | → 01 | 49 → **99** |
| **average** | 42 → 71 | 50 → 69 | 16 → 20 | 44 → 45 | 00 → 02 | 42 → 29 | → 24 | → 23 | → 04 | 44 → **90** |
| **improvement** | + 71.0% | + 37.7% | + 23.7% | + 3.0% | N/A | - 30.3% | N/A | N/A | N/A | **+ 106.9%** |

a

Table 1: **Normalized score before & after online fine-tuning.** Observe that Cal-QL improves over the best prior fine-tuning method and attains a much larger performance improvement over the course of online fine-tuning. The numbers represent the normalized score out of 100 following the convention in [10].

## 7.2  Cal-QL With High Update-to-Data (UTD) Ratio

We can further enhance the online sample efficiency of Cal-QL by increasing the number of gradient steps per environment step made by the algorithm. The number of updates per environment step is usually called the update-to-data (UTD) ratio. In standard online RL, running off-policy Q-learning with a high UTD value (e.g., 20, compared to the typical value of 1) often results in challenges pertaining to overfitting [39, 5, 2, 8]. As expected, we noticed that running Cal-QL with a high UTD value also leads these overfitting challenges. To address these challenges in high UTD settings, we combine Cal-QL with the Q-function architecture in recent work, RLPD [2] (i.e., we utilized layer normalization in the Q-function and ensembles akin to Chen et al. [5]), that attempts to tackle

| Task | CQL | IQL | AWAC | O3F | ODT | CQL+SAC | Hybrid RL | SAC+od | SAC | Cal-QL (Ours) |
|---|---|---|---|---|---|---|---|---|---|---|
| large-diverse | 0.35 | 0.46 | 1.00 | 0.62 | 0.98 | 0.99 | 1.00 | 1.00 | 1.00 | **0.20** |
| large-play | 0.32 | 0.52 | 1.00 | 0.91 | 1.00 | 0.99 | 1.00 | 1.00 | 1.00 | **0.28** |
| medium-diverse | 0.06 | 0.08 | 0.99 | **0.03** | 0.95 | 0.06 | 0.98 | 0.77 | 1.00 | 0.05 |
| medium-play | 0.09 | 0.10 | 0.99 | **0.04** | 0.96 | 0.06 | 0.90 | 0.47 | 1.00 | 0.07 |
| partial | 0.31 | 0.49 | 0.89 | 0.78 | - | 0.97 | 0.98 | 0.98 | 0.92 | **0.27** |
| mixed | 0.55 | 0.60 | 0.88 | 0.72 | - | 0.97 | 0.99 | 1.00 | 0.91 | **0.27** |
| complete | 0.70 | 0.53 | 0.97 | 0.66 | - | 0.99 | 0.99 | 0.96 | 0.91 | **0.44** |
| pen | 0.86 | **0.11** | 0.12 | 0.13 | - | 0.90 | 0.56 | 0.75 | 0.87 | **0.11** |
| door | 0.36 | 0.25 | 0.81 | 0.82 | - | **0.23** | 0.35 | 0.60 | 0.94 | 0.23 |
| relocate | 0.71 | 0.74 | 0.95 | 0.71 | - | 0.86 | **0.30** | 0.89 | 1.00 | 0.43 |
| manipulation | 0.15 | 0.32 | 0.38 | - | - | 0.61 | 1.00 | 1.00 | 0.99 | **0.11** |
| **average** | 0.41 | 0.38 | 0.82 | 0.54 | 0.97 | 0.69 | 0.82 | 0.86 | 0.96 | **0.22** |

Table 2: **Cumulative regret averaged over the steps of fine-tuning.** The smaller the better and 1.00 is the worst. Cal-QL attains the smallest overall regret, achieving the best performance among 8 / 11 tasks.

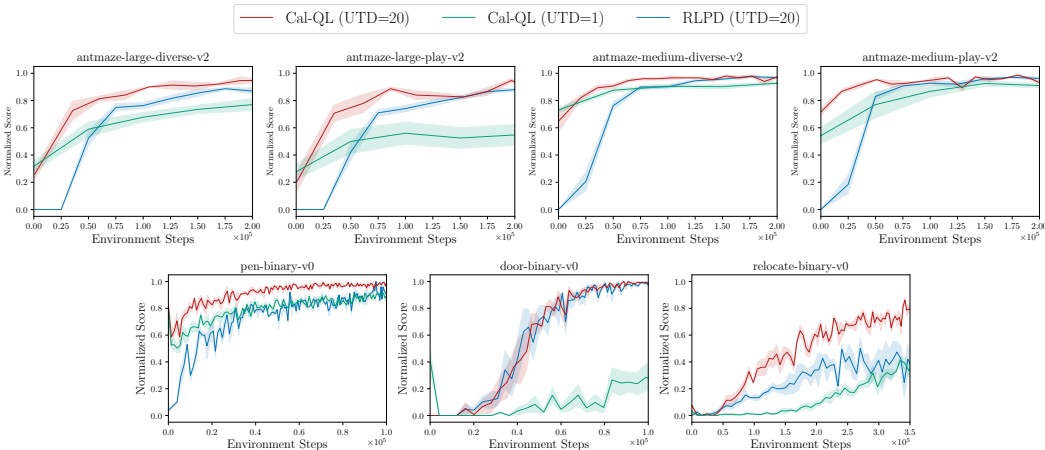

Figure 7: **Cal-QL with UTD=20**. Incorporating design choices from RLPD enables Cal-QL to achieve sample-efficient fine-tuning with UTD=20. Specifically, Cal-QL generally attains similar or higher asymptotic performance as RLPD, while also exhibiting a smaller cumulative regret. (3 seeds)

overfitting challenges. Note that Cal-QL still first pre-trains on the offline dataset using Equation 5.1 followed by online fine-tuning, unlike RLPD that runs online RL right from the start. In Figure 7, we compare Cal-QL (UTD = 20) with RLPD [2] (UTD = 20) and also Cal-QL (UTD = 1) as a baseline. Observe that Cal-QL (UTD = 20) improves over Cal-QL (UTD = 1) and training from scratch (RLPD).

## 7.3 Understanding the Behavior of Cal-QL

In this section, we aim to understand the behavior of Cal-QL by performing controlled experiments that modify the dataset composition, and by investigating various metrics to understand the properties of scenarios where utilizing Cal-QL is especially important for online fine-tuning.

**Effect of data composition.** To understand the efficacy of Cal-QL with different data compositions, we ran it on a newly constructed fine-tuning task on the medium-size AntMaze domain with a low-coverage offline dataset, which is generated via a scripted controller that starts from a fixed initial position and navigates the ant to a fixed goal position. In Figure 8, we plot the performance of Cal-QL and baseline CQL (for comparison) on this task, alongside the trend of average Q-values over the course of offline pre-training (to the left of the dashed vertical line, before 250 training epochs) and online fine-tuning (to the right of the vertical dashed line, after 250 training epochs), and the trend of *bounding rate*, i.e., the fraction of transitions in the data-buffer for which the constraint in Cal-QL actively lower-bounds the learned Q-function with the reference value. For comparison, we also plot these quantities for a diverse dataset with high coverage on the task (we use the antmaze-medium-diverse from Fu et al. [10] as a representative diverse dataset) in Figure 8.

Observe that for the diverse dataset, both naïve CQL and Cal-QL perform similarly, and indeed, the learned Q-values behave similarly for both of these methods. In this setting, online learning doesn't spend samples to correct the Q-function when fine-tuning begins leading to a low bounding rate, almost always close to 0. Instead, with the narrow dataset, we observe that the Q-values learned by

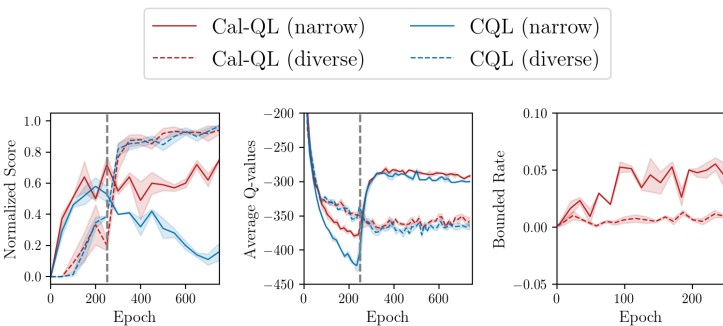

Figure 8: **Performance of Cal-QL with data compositions.** Cal-QL is most effective with narrow datasets, where Q-values need to be corrected at the beginning of fine-tuning.

naïve CQL are much smaller, and are corrected once fine-tuning begins. This correction co-occurs with a drop in performance (solid blue line on left), and naïve CQL is unable to recover from this drop. Cal-QL which calibrates the scale of the Q-function for many more samples in the dataset, stably transitions to online fine-tuning with no unlearning (solid red line on left).

This suggests that in settings with narrow datasets (e.g., in the experiment above and in the `adroit` and `visual-manipulation` domains from Figure 6), Q-values learned by naïve conservative methods are more likely to be smaller than the ground-truth Q-function of the behavior policy due to function approximation errors. Hence utilizing Cal-QL to calibrate the Q-function against the behavior policy can be significantly helpful. On the other hand, with significantly high-coverage datasets, especially in problems where the behavior policy is also random and sub-optimal, Q-values learned by naïve methods are likely to already be calibrated with respect to those of the behavior policy. Therefore no explicit calibration might be needed (and indeed, the bounding rate tends to be very close to 0 as shown in Figure 8). In this case, Cal-QL will revert back to standard CQL, as we observe in the case of the diverse dataset above. This intuition is also reflected in Theorem 6.1: when the reference policy $\mu$ is close to a narrow, expert policy, we would expect Cal-QL to be especially effective in controlling the efficiency of online fine-tuning.

**Estimation errors in the reference value function do not affect performance significantly.** In our experiments, we compute the reference value functions using Monte-Carlo return estimates. However, this may not be available in all tasks. How does Cal-QL behave when reference value functions must be estimated using the offline dataset itself? To answer this, we ran an experiment on the `kitchen` domain, where instead of using an estimate for $Q^\mu$ based on the Monte-Carlo return, we train a neural network function approximator $Q_\theta^\mu$ to approximate $Q^\mu$ via supervised regression on to Monte-Carlo return, which is then utilized by Cal-QL. Observe in Figure 9, that the performance of Cal-QL largely remains unaltered. This implies as long as we can obtain a reasonable function approximator to estimate the Q-function of the reference policy (in this case, the behavior policy), errors in the reference Q-function do not affect the performance of Cal-QL significantly.

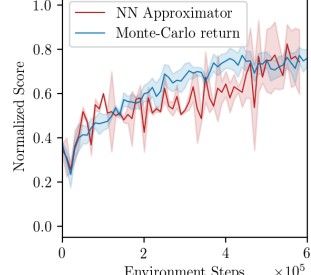

Figure 9: **Using a neural network approximator for the reference value function performs comparable to using the Monte-Carlo return.** This indicates that errors in the reference Q-function do not negatively impact the performance.

## 8    Discussion, Future Directions, and Limitations

In this work we developed Cal-QL a method for acquiring conservative offline initializations that facilitate fast online fine-tuning. Cal-QL learns conservative value functions that are constrained to be larger than the value function of a reference policy. This form of calibration allows us to avoid initial unlearning when fine-tuning with conservative methods, while also retaining the effective asymptotic performance that these methods exhibit. Our theoretical and experimental results highlight the benefit of Cal-QL in enabling fast online fine-tuning. While Cal-QL outperforms prior methods, we believe that we can develop even more effective methods by adjusting calibration and conservatism more carefully. A limitation of our work is that we do not consider fine-tuning setups where pre-training and fine-tuning tasks are different, but this is an interesting avenue for future work.

## Acknowledgments

This research was partially supported by the Office of Naval Research N00014-21-1-2838, N00014-22-1-2102, ARO W911NF-21-1-0097, the joint Simons Foundation-NSF DMS grant #2031899, AFOSR FA9550-22-1-0273, and Tsinghua-Berkeley Shenzhen Institute (TBSI) Research Fund, as well as support from Intel and C3.ai, the Savio computational cluster resource provided by the Berkeley Research Computing program, and computing support from Google. We thank Philip J. Ball, Laura Smith, and Ilya Kostrikov for sharing the experimental results of RLPD. AK is supported by the Apple Scholars in AI/ML PhD Fellowship. MN is partially supported by the Nakajima Foundation Fellowship. YZ is partially supported by Siemens CITRIS and TBSI research fund. YZ would like to thank Prof. Song Mei for insightful suggestions on the presentation of Theorem 6.1.

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

# Appendices

## A Implementation details of Cal-QL

Our algorithm, Cal-QL is illustrated in Algorithm 1. A python-style implementation is provided in Appendix A.2. The official code and experimental logs are available at https://github.com/nakamotoo/Cal-QL

### A.1 Cal-QL Algorithm

We use $J_Q(\theta)$ to denote the calibrated conservative regularizer for the Q network update:

$$J_Q(\theta) := \alpha \underbrace{\left( \mathbb{E}_{s \sim \mathcal{D}, a \sim \pi} \left[ \max \left( Q_\theta(s, a), Q^\mu(s, a) \right) \right] - \mathbb{E}_{s, a \sim \mathcal{D}} \left[ Q_\theta(s, a) \right] \right)}_{\text{Calibrated conservative regularizer } \mathcal{R}(\theta)} \tag{A.1}$$

$$+ \frac{1}{2} \mathbb{E}_{s, a, s' \sim \mathcal{D}} \left[ \left( Q_\theta(s, a) - \mathcal{B}^\pi \bar{Q}(s, a) \right)^2 \right]. \tag{A.2}$$

---

**Algorithm 1** Cal-QL pseudo-code

1: Initialize Q-function, $Q_\theta$, a policy, $\pi_\phi$
2: **for** step $t$ in $\{1, \ldots, N\}$ **do**
3:     Train the Q-function using the conservative regularizer in Eq. A.1:

$$\theta_t := \theta_{t-1} - \eta_Q \nabla_\theta J_Q(\theta) \tag{A.3}$$

4:     Improve policy $\pi_\phi$ with SAC-style update:

$$\phi_t := \phi_{t-1} + \eta_\pi \mathbb{E}_{s \sim \mathcal{D}, a \sim \pi_\phi(\cdot|s)} [Q_\theta(s, a) - \log \pi_\phi(a|s)] \tag{A.4}$$

5: **end for**

---

### A.2 Python Implementation

Listing 1: Training Q networks given a batch of data

```python
q_data = critic(batch['observations'], batch['actions'])

next_dist = actor(batch['next_observations'])
next_pi_actions, next_log_pis = next_dist.sample()

target_qval = target_critic(batch['observations'], next_pi_actions)
target_qval = batch['rewards'] + self.gamma * (1 - batch['dones']) * target_qval

td_loss = mse_loss(q_data, target_qval)

num_samples = 4
random_actions = uniform((num_samples, batch_size, action_dim), min=-1, max=1)
random_pi = 0.5 ** batch['actions'].shape[-1]

pi_actions, log_pis = actor(batch['observations'])

q_rand_is = critic(batch['observations'], random_actions) - random_pi
q_pi_is = critic(batch['observations'], pi_actions) - log_pis

# Cal-QL's modification
mc_return = batch['mc_return'].repeat(num_samples)
q_pi_is = max(q_pi_is, mc_return)

cat_q = concatenate([q_rand_is, q_pi_is], new_axis=True)
cat_q = logsumexp(cat_q, axis=0) # sum over num_samples
critic_loss = td_loss + ((cat_q - q_data).mean() * cql_alpha)

critic_optimizer.zero_grad()
critic_loss.backward()
critic_optimizer.step()
```

Listing 2: Training the policy (or the actor) given a batch of data

```
# return distribution of actions
pi_actions, log_pis = actor(batch['observations'])

# calculate q value of actor actions
qpi = critic(batch['observations'], actions)
qpi = qpi.min(axis=0)

# same objective as CQL (kumar et al.)
actor_loss = (log_pis * self.alpha - qpi).mean()

# optimize loss
actor_optimizer.zero_grad()
actor_loss.backward()
actor_optimizer.step()
```

# B    Environment Details

## B.1    Antmaze

The Antmaze navigation tasks aim to control an 8-DoF ant quadruped robot to move from a starting point to a desired goal in a maze. The agent will receive sparse +1/0 rewards depending on whether it reaches the goal or not. We study each method on "medium" and "hard" (shown in Figure 5) mazes which are difficult to solve, using the following datasets from D4RL [10]: `large-diverse`, `large-play`, `medium-diverse`, and `medium-play`. The difference between "diverse" and "play" datasets is the optimality of the trajectories they contain. The "diverse" datasets contain the trajectories commanded to a random goal from random starting points, while the "play" datasets contain the trajectories commanded to specific locations which are not necessarily the goal. We used an episode length of 1000 for each task. For Cal-QL, CQL, and IQL, we pre-trained the agent using the offline dataset for 1M steps. We then trained online fine-tuning for 1M environment steps for each method.

## B.2    Franka Kitchen

The Franka Kitchen domain require controlling a 9-DoF Franka robot to arrange a kitchen environment into a desired configuration. The configuration is decomposed into 4 subtasks, and the agent will receive rewards of $0, +1, +2, +3$, or $+4$ depending on how many subtasks it has managed to solve. To solve the whole task and reach the desired configuration, it is important to learn not only how to solve each subtask, but also to figure out the correct order to solve. We study this domain using datasets with three different optimalities: `kitchen-complete`, `kitchen-partial`, and `kitchen-mixed`. The "complete" dataset contains the trajectories of the robot performing the whole task completely. The "partial" dataset partially contains some complete demonstrations, while others are incomplete demonstrations solving the subtasks. The "mixed" dataset only contains incomplete data without any complete demonstrations, which is hard and requires the highest degree of stitching and generalization ability. We used an episode length of 1000 for each task. For Cal-QL, CQL, and IQL, we pre-trained the agent using the offline dataset for 500K steps. We then performed online fine-tuning for 1.25M environment steps for each method.

## B.3    Adroit

The Adroit domain involves controlling a 24-DoF shadow hand robot. There are 3 tasks we consider in this domain: `pen-binary`, `relocate-binary`, `relocate-binary`. These tasks comprise a limited set of narrow human expert data distributions ($\sim 25$) with additional trajectories collected by a behavior-cloned policy. We truncated each trajectory and used the positive segments (terminate when the positive reward signal is found) for all methods. This domain has a very narrow dataset distribution and a large action space. In addition, learning in this domain is difficult due to the sparse reward, which leads to exploration challenges. We utilized a variant of the dataset used in prior work [44] to have a standard comparison with SOTA offline fine-tuning experiments that consider this domain. For the offline learning phase, we pre-trained the agent for 20K steps. We then performed online fine-tuning for 300K environment steps for the `pen-binary` task, and 1M environment steps for the `door-binary` and `relocate-binary` tasks. The episode length is 100, 200, and 200 for `pen-binary`, `door-binary`, and `relocate-binary` respectively.

### B.4 Visual Manipulation Domain

The Visual Manipulation domain consists of a pick-and-place task. This task is a multitask formulation explored in the work, Pre-training for Robots (PTR) [33]. Here each task is defined as placing an object in a bin. A distractor object was present in the scene as an adversarial object which the agent had to avoid picking. There were 10 unique objects and no overlap between the task objects and the interfering/distractor objects. The episode length is 40. For the offline phase, we pre-trained the policy with offline data for 50K steps. We then performed online fine-tuning for 100K environment steps for each method, taking 5 gradient steps per environment step.

## C Experiment Details

### C.1 Normalized Scores

The `visual-manipulation`, `adroit`, and `antmaze` domains are all goal-oriented, sparse reward tasks. In these domains, we computed the normalized metric as simply the goal achieved rate for each method. For example, in the visual manipulation environment, if the object was placed successfully in the bin, a +1 reward was given to the agent and the task is completed. Similarly, for the `door-binary` task in the adroit tasks, we considered the success rate of opening the door. For the `kitchen` task, the task is to solve a series of 4 sub-tasks that need to be solved in an iterative manner. The normalized score is computed simply as $\frac{\#\text{tasks solved}}{\text{total tasks}}$.

### C.2 Mixing Ratio Hyperparameter

In this work, we explore the mixing ratio parameter $m$, which is used during the online fine-tuning phase. The mixing ratio is either a value in the range $[0, 1]$ or the value -1. If this mixing ratio is within $[0, 1]$, it represents what percentage of offline and online data is seen in each batch when fine-tuning. For example, if the mixing ratio $m = 0.25$, that means for each batch we sample 25% from the offline data and 75% from online data. Instead, if the mixing ratio is -1, the buffers are appended to each other and sampled uniformly. We present an ablation study over mixing ratio in Appendix G.

### C.3 Details and Hyperparameters for CQL and Cal-QL

We list the hyperparameters for CQL and Cal-QL in Table 3. We utilized a variant of Bellman backup that computes the target value by performing a maximization over target values computed for $k$ actions sampled from the policy at the next state, where we used $k = 4$ in visual pick and place domain and $k = 10$ in others. In the Antmaze domain, we used the dual version of CQL [32] and conducted ablations over the value of the threshold of the CQL regularizer $\mathcal{R}(\theta)$ (target action gap) instead of $\alpha$. In the visual-manipulation domain which is not presented in the original paper, we swept over the alpha values of $\alpha = 0.5, 1, 5, 10$, and utilized separate $\alpha$ values for offline ($\alpha = 5$) and online ($\alpha = 0.5$) phases for the final results. We built our code upon the CQL implementation from https://github.com/young-geng/JaxCQL [14]. We used a single NVIDIA TITAN RTX chip to run each of our experiments.

### C.4 Details and Hyperparameters for IQL

We list the hyperparameters for IQL in Table 4. To conduct our experiments, we used the official implementation of IQL provided by the authors [30], and primarily followed their recommended parameters, which they previously ablated over in their work. In the visual-manipulation domain which is not presented in the original paper, we performed a parameter sweep over expectile $\tau = 0.5, 0.6, 0.7, 0.8, 0.9, 0.95, 0.99$ and temperature $\beta = 1, 3, 5, 10, 25, 50$ and selected the best-performing values of $\tau = 0.7$ and $\beta = 10$ for our final results. In addition, as the second best-performing method in the visual-manipulation domain, we also attempted to use separate $\beta$ values for IQL, for a fair comparison with CQL and Cal-QL. However, we found that it has little to no effect, as shown in Figure 10.

### C.5 Details and Hyperparameters for AWAC and ODT

We used the JAX implementation of AWAC from https://github.com/ikostrikov/jaxrl [29]. We primarily followed the author's recommended parameters, where we used the Lagrange multiplier

$\lambda = 1.0$ for the Antmaze and Franka Kitchen domains, and $\lambda = 0.3$ for the Adroit domain. In the visual-manipulation domain, we performed a parameter sweep over $\lambda = 0.1, 0.3, 1, 3, 10$ and selected the best-performing value of $\lambda = 1$ for our final results. For ODT, we used the author's official implementation from https://github.com/facebookresearch/online-dt, with the author's recommended parameters they used in the Antmaze domain. In addition, in support of our result of AWAC and ODT (as shown in Table 1), the poor performance of Decision Transformers and AWAC in the Antmaze domain can also be observed in Table 1 and Table 2 of the IQL paper [30].

## C.6 Details and Hyperparameters for SAC, SAC + Offline Data, Hybrid RL and CQL + SAC

We used the standard hyperparameters for SAC as derived from the original implementation in [19]. We used the same other hyperparameters as CQL and Cal-QL. We used automatic entropy tuning for the policy and critic entropy terms, with a target entropy of the negative action dimension. For SAC, the agent is only trained with the online explored data. For SAC + Offline Data, the offline data and online explored data is combined together and sampled uniformly. For Hybrid RL, we use the same mixing ratio used for CQL and Cal-QL presented in Table 3. For CQL + SAC, we first pre-train with CQL and then run online fine-tuning using both offline and online data, also using the same mixing ratio presented in Table 3.

Table 3: CQL, Cal-QL Hyperparameters

| Hyperparameters | Adroit | Kitchen | Antmaze | Manipulation |
|---|---|---|---|---|
| $\alpha$ | 1 | 5 | - | 5 (online: 0.5) |
| target action gap | - | - | 0.8 | - |
| mixing ratio | -1, 0.25, **0.5** | -1, **0.25**, 0.5 | 0.5 | 0.2, **0.5**, 0.7, 0.9 |

Table 4: IQL Hyperparameters

| Hyperparameters | Adroit | Kitchen | Antmaze | Manipulation |
|---|---|---|---|---|
| expectile $\tau$ | 0.8 | 0.7 | 0.9 | 0.7 |
| temperature $\beta$ | 3 | 0.5 | 10 | 10 |
| mixing ratio | -1, **0.2**, 0.5 | -1, **0.25**, 0.5 | 0.5 | 0.2, **0.5**, 0.7, 0.9 |

## D D4RL locomotion benchmark

In this section, we further compare Cal-QL with previous methods on the D4RL locomotion benchmarks. Since the dataset for these tasks does not end in a terminal, we estimate the reference value functions using a neural network by fitting a SARSA Q-function to the offline dataset. We present the normalized scores before and after fine-tuning in Table 5, and the cumulative regret in Table 6. Overall, we observe that Cal-QL outperforms state-of-the-art methods such as IQL and AWAC, as well as fast online RL methods that do not employ pre-training (Hybrid RL), performing comparably to CQL.

| Task | CQL | IQL | AWAC | Hybrid RL | Cal-QL (Ours) |
|---|---|---|---|---|---|
| halfcheetah-expert-v2 | 0.79 → 1.02 | 0.95 → 0.54 | 0.69 → 0.65 | N/A → 0.84 | 0.85 → 1.00 |
| halfcheetah-medium-expert-v2 | 0.59 → 1.00 | 0.86 → 0.57 | 0.72 → 0.77 | N/A → 0.86 | 0.54 → 0.99 |
| halfcheetah-medium-replay-v2 | 0.51 → 0.95 | 0.43 → 0.36 | 0.43 → 0.47 | N/A → 0.89 | 0.51 → 0.93 |
| halfcheetah-medium-v2 | 0.53 → 0.97 | 0.47 → 0.43 | 0.49 → 0.57 | N/A → 0.88 | 0.52 → 0.93 |
| halfcheetah-random-v2 | 0.36 → 1.01 | 0.11 → 0.54 | 0.14 → 0.37 | N/A → 0.80 | 0.33 → 1.04 |
| hopper-expert-v2 | 1.00 → 0.73 | 1.02 → 0.52 | 1.12 → 1.11 | N/A → 0.54 | 0.58 → 0.75 |
| hopper-medium-expert-v2 | 0.86 → 0.92 | 0.07 → 0.81 | 0.30 → 1.03 | N/A → 1.00 | 0.69 → 0.76 |
| hopper-medium-replay-v2 | 0.69 → 1.11 | 0.76 → 0.86 | 0.70 → 1.08 | N/A → 0.77 | 0.76 → 1.10 |
| hopper-medium-v2 | 0.78 → 0.99 | 0.57 → 0.26 | 0.58 → 0.90 | N/A → 1.06 | 0.89 → 0.98 |
| hopper-random-v2 | 0.09 → 1.08 | 0.08 → 0.64 | 0.09 → 0.43 | N/A → 0.80 | 0.10 → 0.79 |
| walker2d-expert-v2 | 1.08 → 1.12 | 1.09 → 1.02 | 1.11 → 1.24 | N/A → 1.12 | 0.92 → 1.00 |
| walker2d-medium-expert-v2 | 1.00 → 0.56 | 1.09 → 1.06 | 0.86 → 1.16 | N/A → 0.95 | 0.96 → 0.77 |
| walker2d-medium-replay-v2 | 0.76 → 0.92 | 0.63 → 0.95 | 0.60 → 0.94 | N/A → 1.03 | 0.52 → 0.99 |
| walker2d-medium-v2 | 0.80 → 1.11 | 0.81 → 1.04 | 0.75 → 0.98 | N/A → 0.86 | 0.75 → 1.03 |
| walker2d-random-v2 | 0.05 → 0.85 | 0.06 → 0.09 | 0.04 → 0.04 | N/A → 0.90 | 0.04 → 0.49 |
| **average** | 0.66 → 0.96 | 0.60 → 0.65 | 0.57 → 0.78 | N/A → 0.89 | 0.60 → 0.90 |

Table 5: Normalized score before & after online fine-tuning on D4RL locomotion tasks.

| Task | CQL | IQL | AWAC | Hybrid RL | Cal-QL (Ours) |
|---|---|---|---|---|---|
| `halfcheetah-expert-v2` | 0.03 | 0.47 | 0.28 | 0.49 | 0.08 |
| `halfcheetah-medium-expert-v2` | 0.05 | 0.42 | 0.24 | 0.49 | 0.07 |
| `halfcheetah-medium-replay-v2` | 0.14 | 0.65 | 0.56 | 0.29 | 0.16 |
| `halfcheetah-medium-v2` | 0.15 | 0.61 | 0.48 | 0.31 | 0.17 |
| `halfcheetah-random-v2` | 0.11 | 0.54 | 0.67 | 0.33 | 0.10 |
| `hopper-expert-v2` | 0.17 | 0.58 | -0.07 | 0.37 | 0.22 |
| `hopper-medium-expert-v2` | 0.09 | 0.32 | -0.02 | 0.32 | 0.17 |
| `hopper-medium-replay-v2` | 0.09 | 0.09 | 0.05 | 0.27 | 0.12 |
| `hopper-medium-v2` | 0.04 | 0.73 | 0.08 | 0.26 | 0.05 |
| `hopper-random-v2` | 0.17 | 0.54 | 0.74 | 0.37 | 0.27 |
| `walker2d-expert-v2` | -0.02 | 0.15 | -0.18 | 0.11 | 0.15 |
| `walker2d-medium-expert-v2` | 0.03 | 0.02 | -0.14 | 0.17 | 0.10 |
| `walker2d-medium-replay-v2` | 0.06 | 0.10 | 0.19 | 0.30 | 0.13 |
| `walker2d-medium-v2` | 0.11 | 0.12 | 0.07 | 0.29 | 0.22 |
| `walker2d-random-v2` | 0.53 | 0.92 | 0.96 | 0.56 | 0.49 |
| **average** | 0.12 | 0.42 | 0.26 | 0.33 | 0.17 |

Table 6: Cumulative regret on D4RL locomotion tasks. The smaller the better and 1.00 is the worst.

# E    Extended Discussion on Limitations of Existing Fine-Tuning Methods

In this section, we aim to highlight some potential reasons behind the slow improvement of other methods in our empirical analysis experiment in Section 4.1, and specifically, we use IQL for the analysis. We first swept over the temperature $\beta$ values used in the online fine-tuning phase for IQL, which controls the constraint on how closely the learned policy should match the behavior policy. As shown in Figure 10, the change in the temperature $\beta$ has little to no effect on the sample efficiency. Another natural hypothesis is that IQL improves slowly because we are not making enough updates per unit of data collected by the environment. To investigate this, we ran IQL with **(a)** five times as many gradient steps per step of data collection (UTD = 5), and **(b)** with a more aggressive policy update. Observe in Figure 11 that **(a)** does not improve the asymptotic performance of IQL, although it does improve CQL meaning that there is room for improvement on this task by making more gradient updates. Observe in Figure 11 that **(b)** often induces policy unlearning, similar to the failure mode in CQL. These two observations together indicate that a policy constraint approach can slow down learning asymptotically, and we cannot increase the speed by making more aggressive updates as this causes the policy to find erroneously optimistic out-of-distribution actions, and unlearn the policy learned from offline data.

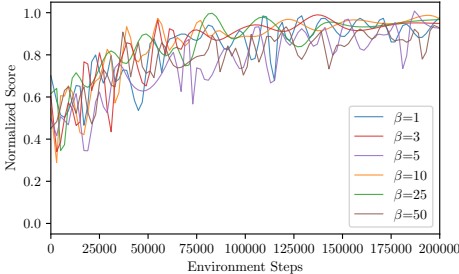

Figure 10: **Abalation on IQL's online temperature values**: The change in the temperature $\beta$ used in online fine-tuning phase has little to no effect on the sample efficiency.

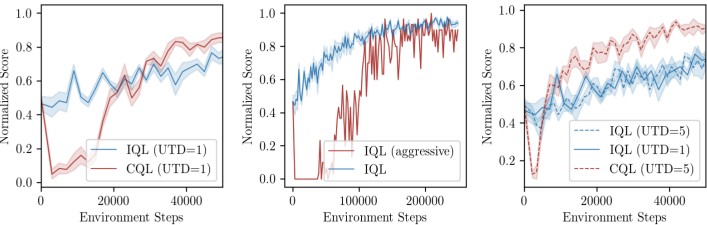

Figure 11: **IQL and CQL:** Step 0 on the x-axis is the performance after offline pre-training. Observe while CQL suffers from initial policy unlearning, IQL improves slower throughout fine-tuning.

## F    Initial Unlearning of CQL on Multiple Tasks

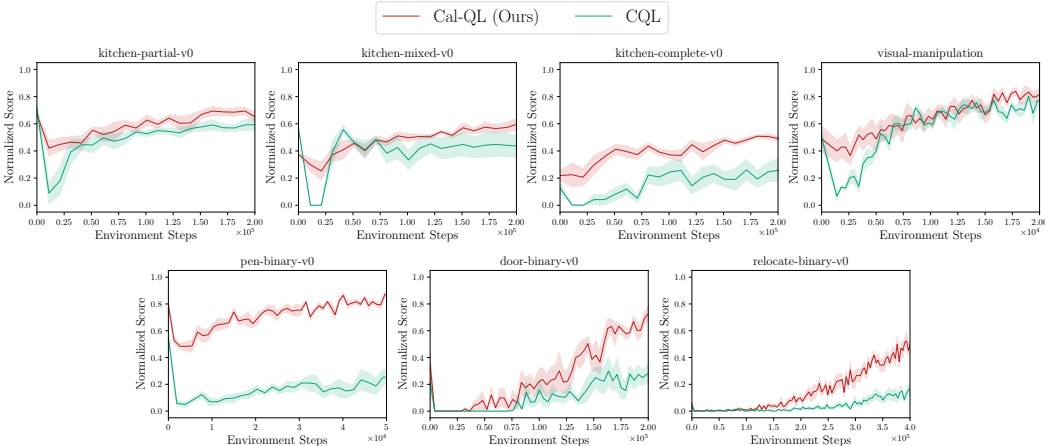

Figure 12: While CQL experiences initial unlearning, Cal-QL effectively mitigates it and quickly recovers its performance.

In this section, we show the learning curves of CQL and Cal-QL from Figure 6 and zoom in on the x-axis to provide a clearer visualization of CQL's initial unlearning in the Franka Kitchen, Adroit, and the visual-manipulation domains. As depicted in Figure 12, it is evident across all tasks that while CQL experiences initial unlearning, Cal-QL can effectively mitigate it and quickly recovers its performance. Regarding the Antmaze domain, as we discussed in section 7.3, CQL does not exhibit initial unlearning since the default dataset has a high coverage of data. However, we can observe a similar phenomenon if we narrow down the dataset distribution (as shown in Figure 8).

## G    Ablation Study over Mixing Ratio Hyperparameter

Here we present an ablation study over mixing ratio on the adroit door-binary task for Cal-QL (left) and IQL (right). Perhaps as intuitively expected, we generally observe the trend that a larger mixing ratio (i.e., more offline data) exhibits slower performance improvement during online fine-tuning.

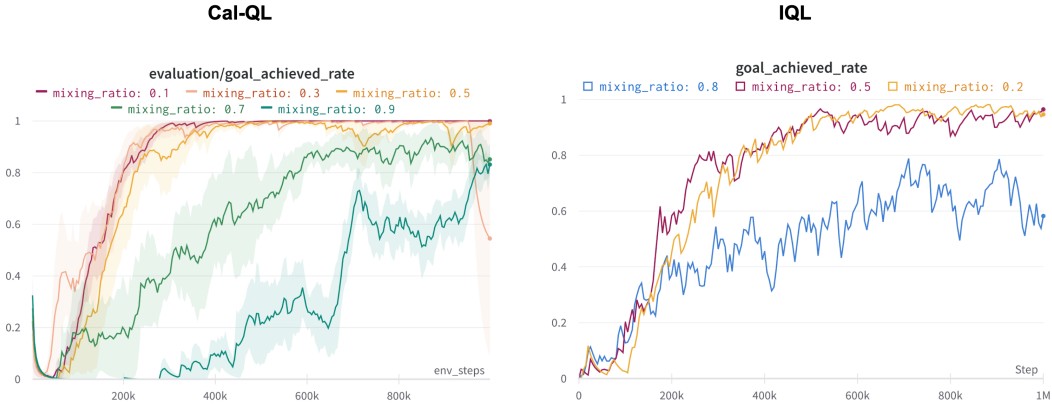

Figure 13: Ablation study over mixing ratio on the adroit door-binary task for Cal-QL (left) and IQL (right).

## H    Regret Analysis of Cal-QL

We provide a theoretical version of Cal-QL in Algorithm 2. Policy fine-tuning has been studied in different settings [60, 52, 55]. Our analysis largely adopts the settings and results in Song et al. [52], with additional changes in Assumption H.1, Assumption H.3, and Definition H.4. Note that the goal of this proof is to demonstrate that *a pessimistic functional class (Assumption H.1)* allows one to utilize the offline data efficiently, rather than providing a new analytical technique for regret analysis.

See comparisons between Section H.3 and Section I.1. Note that we use $f$ instead of $Q_\theta$ in the main text to denote the estimated $Q$ function for notation simplicity.

---

**Algorithm 2** Theoretical version of Cal-QL

---

1: **Input:** Value function class $\mathcal{F}$, # total iterations $K$, offline dataset $\mathcal{D}_h^\nu$ of size $m_{\text{off}}$ for $h \in [H-1]$.
2: Initialize $f_h^1(s,a) = 0, \forall(s,a)$.
3: **for** $k = 1, \ldots, K$ **do**
4:     Let $\pi^t$ be the greedy policy w.r.t. $f^k$             ▷ I.e., $\pi_h^k(s) = \arg\max_a f_h^k(s,a)$.
5:     For each $h$, collect $m_{\text{on}}$ online tuples $\mathcal{D}_h^k \sim d_h^{\pi^k}$           ▷ online data collection
6:     Set $f_H^{k+1}(s,a) = 0, \forall(s,a)$.
7:     **for** $h = H-1, \ldots 0$ **do**             ▷ FQI with offline and online data
8:         Estimate $f_h^{k+1}$ using conservative least squares on the aggregated data:    ▷ I.e., CQL regularized class $\mathcal{C}_h$

$$f_h^{k+1} \leftarrow \arg\min_{f \in \mathcal{C}_h} \left\{ \widehat{\mathbb{E}}_{\mathcal{D}_h^\nu} \left[ f(s,a) - r - \max_{a'} f_{h+1}^{k+1}(s',a') \right]^2 + \sum_{\tau=1}^K \widehat{\mathbb{E}}_{\mathcal{D}_h^\tau} \left[ f(s,a) - r - \max_{a'} f_{h+1}^{k+1}(s',a') \right]^2 \right\}$$
(H.1)

9:         $f_h^{k+1} = \max\{f_h^{k+1}, Q_h^{\text{ref}}\}$       ▷ Set a reference policy for calibration (Definition 4.1)
10:     **end for**
11: **end for**
12: **Output:** $\pi^K$

---

## H.1 Preliminaries

In this subsection, we follow most of the notations and definitions in Song et al. [52]. In particular, we consider the finite horizon cases, where the value function and Q function are defined as:

$$V_h^\pi(s) = \mathbb{E}\left[ \sum_{\tau=h}^{H-1} r_\tau | \pi, s_h = s \right] \tag{H.2}$$

$$Q_h^\pi(s,a) = \mathbb{E}\left[ \sum_{\tau=h}^{H-1} r_\tau | \pi, s_h = s, a_h = a \right]. \tag{H.3}$$

We also define the Bellman operator $\mathcal{T}$ such that $\forall f : \mathcal{S} \times \mathcal{A}$:

$$\mathcal{T}f(s,a) = \mathbb{E}_{s,a}[R(s,a)] + \mathbb{E}_{s' \sim P(s,a)} \max_{a'} f(s',a'), \ \forall(s,a) \in \mathcal{S} \times \mathcal{A}, \tag{H.4}$$

where $R(s,a) \in \Delta[0,1]$ represents a stochastic reward function.

## H.2 Notations

- Feature covariance matrix $\Sigma_{k;h}$:

$$\mathbf{\Sigma}_{k;h} = \sum_{\tau=1}^k X_h(f^\tau)(X_h(f^\tau))^\top + \lambda \mathbf{I} \tag{H.5}$$

- Matrix Norm Zanette et al. [63]: for a matrix $\Sigma$, the matrix norm $\|\boldsymbol{u}\|_{\mathbf{\Sigma}}$ is defined as:

$$\|\boldsymbol{u}\|_{\mathbf{\Sigma}} = \sqrt{\boldsymbol{u}\mathbf{\Sigma}\boldsymbol{u}^\top} \tag{H.6}$$

- Weighted $\ell^2$ norm: for a given distribution $\beta \in \Delta(\mathcal{S} \times \mathcal{A})$ and a function $f : \mathcal{S} \times \mathcal{A} \mapsto \mathbb{R}$, we denote the weighted $\ell^2$ norm as:

$$\|f\|_{2,\beta}^2 := \sqrt{\mathbb{E}_{(s,a) \sim \beta} f^2(s,a)} \tag{H.7}$$

- A stochastic reward function $R(s,a) \in \Delta([0,1])$
- For each offline data distribution $\nu = \{\nu_0, \ldots, \nu_{H-1}\}$, the offline data set at time step $h$ ($\nu_h$) contains data samples $(s,a,r,s')$, where $(s,a) \sim \nu_h, r \in R(s,a), s' \sim P(s,a)$.

- Given a policy $\pi := \{\pi_0, \ldots, \pi_{H-1}\}$, where $\pi_h : \mathcal{S} \mapsto \Delta(\mathcal{A})$, $d_h^\pi \in \Delta(s, a)$ denotes the state-action occupancy induced by $\pi$ at step $h$.

- We consider the value-based function approximation setting, where we are given a function class $\mathcal{C} = \mathcal{C}_0 \times \ldots \mathcal{C}_{H-1}$ with $\mathcal{C}_h \subset \mathcal{S} \times \mathcal{A} \mapsto [0, V_{\max}]$.

- A policy $\pi^f$ is defined as the greedy policy w.r.t. $f$: $\pi_h^f(s) = \arg\max_a f_h(s, a)$. Specifically, at iteration $k$, we use $\pi^k$ to denote the greedy policy w.r.t. $f^k$.

## H.3 Assumptions and Defintions

**Assumption H.1** (Pessimistic Realizability and Completeness). *For any policy $\pi^e$, we say $\mathcal{C}_h$ is a pessimistic function class w.r.t. $\pi^e$, if for any $h$, we have $Q_h^{\pi^e} \in \mathcal{C}_h$, and additionally, for any $f_{h+1} \in \mathcal{C}_{h+1}$, we have $\mathcal{T} f_{h+1} \in \mathcal{C}_h$ and $f_h(s, a) \le Q_h^{\pi^e}(s, a), \forall (s, a) \in \mathcal{S} \times \mathcal{A}$.*

**Definition H.2** (Bilinear model Du et al. [9]). *We say that the MDP together with the function class $\mathcal{F}$ is a bilinear model of rand d of for any $h \in [H - 1]$, there exist two (known) mappings $X_h, W_h : \mathcal{F} \mapsto \mathbb{R}^d$ with $\max_f \|X_h(f)\|_2 \le B_X$ and $\max_f \|W_h(f)\|_2 \le B_W$ such that*

$$\forall f, g \in \mathcal{F}: \quad \left| \mathbb{E}_{s, a \sim d_h^{\pi^f}} [g_h(s, a) - \mathcal{T} g_{h+1}(s, a)] \right| = |\langle X_h(f), W_h(g) \rangle|. \tag{H.8}$$

**Assumption H.3** (Bilinear Rank of Reference Policies). *Suppose $Q^{\mathrm{ref}} \in \mathcal{C}_{\mathrm{ref}} \subset \mathcal{C}$, where $\mathcal{C}_{\mathrm{ref}}$ is the function class of our reference policy, we assume the Bilinear rank of $\mathcal{C}_{\mathrm{ref}}$ is $d_{\mathrm{ref}}$ and $d_{\mathrm{ref}} \le d$.*

**Definition H.4** (Calibrated Bellman error transfer coefficient). *For any policy $\pi$, we define the calibrated transfer coefficient w.r.t. to a reference policy $\pi^{\mathrm{ref}}$ as*

$$C_\pi^{\mathrm{ref}} := \max_{f \in \mathcal{C}, f(s,a) \ge Q^{\mathrm{ref}}(s,a)} \frac{\sum_{h=0}^{H-1} \mathbb{E}_{s, a \sim d_h^\pi} [\mathcal{T} f_{h+1}(s, a) - f_h(s, a)]}{\sqrt{\sum_{h=0}^{H-1} \mathbb{E}_{s, a \sim \nu_h} (\mathcal{T} f_{h+1}(s, a) - f_h(s, a))^2}}, \tag{H.9}$$

*where $Q^{\mathrm{ref}} = Q^{\pi^{\mathrm{ref}}}$.*

## H.4 Discussions on the Assumptions

The pessimistic realizability and completeness assumption (Assumption H.1) is motivated by some theoretical studies of the pessimistic offline methods [59, 6] with regularizers:

$$\min_\theta \alpha \underbrace{\left( \mathbb{E}_{s \sim \mathcal{D}, a \sim \pi} [Q_\theta(s, a)] - \mathbb{E}_{s, a \sim \mathcal{D}} [Q_\theta(s, a)] \right)}_{\text{Conservative regularizer } \mathcal{R}(\theta)} + \frac{1}{2} \mathbb{E}_{s, a, s' \sim \mathcal{D}} \left[ \left( Q_\theta(s, a) - \mathcal{B}^\pi \bar{Q}(s, a) \right)^2 \right].$$

$$\tag{H.10}$$

Since the goal of the conservative regularizer $\mathcal{R}(\theta)$ intrinsically wants to enforce

$$Q_\theta(s, \pi(s)) \le Q_\theta(s, \pi^e(s)), \tag{H.11}$$

where $\pi$ is the training policy and $\pi^e$ is the reference (behavior) policy. One can consider (H.10) as the Lagrange duality formulation of the following primal optimization problem:

$$\min_\theta \mathbb{E}_{s, a, s' \sim \mathcal{D}} \left[ \left( Q_\theta(s, a) - \mathcal{B}^\pi \bar{Q}(s, a) \right)^2 \right], \text{ subject to } \mathbb{E}_{s \sim \mathcal{D}, a \sim \pi} [Q_\theta(s, a)] \le \mathbb{E}_{s \sim \mathcal{D}, a \sim \pi^e} [Q_\theta(s, a)],$$

$$\tag{H.12}$$

where the constraint set is equivalent to Assumption H.1. Although Assumption H.1 directly characterizes the constraint set of the primal form of (H.10) the exact theoretical connection between the pessimistic realizability and the regularized bellman consistency equation is beyond the scope of this work and we would like to leave that for future studies.

Assumption H.1 allows us to restrict the functional class of interest to a smaller conservative function class $\mathcal{C} \subset \mathcal{F}$, which leads to a smaller Bellman rank of the reference policy ($d_{\mathrm{ref}} \le d$) suggested in Assumption H.3, and a smaller concentrability coefficient ($C_\pi^{\mathrm{ref}} \le C_\pi$) defined in Definition H.4, and I.2. Assumption H.3 and Definition I.2 provide the Bellman Bilinear rank and Bellman error transfer coefficient of the pessimistic functional class $\mathcal{C}$ of interest.

## H.5 Proof Structure Overview

We provide an overview of the proof structure and its dependency on different assumptions below:

- Theorem H.5: the total regret is decomposed into *offline regrets* and *online regrets*.
  - Bounding *offline regrets*, requiring Definition H.4 and the following lemmas:
    * Performance difference lemma w.r.t. a comparator policy (Lemma I.5).
    * Least square generalization bound (Lemma I.4), requiring Assumption H.1.
  - Bounding *online regrets*, requiring Definition H.2
    * Performance difference lemma for the online error (Lemma I.6).
    * Least square generalization bound (Lemma I.4), requiring Assumption H.1.
    * Upper bounds with the bilinear model assumption (Lemma I.7).
    * Applying Elliptical Potential Lemma [35] with bellman rank $d$ and $d_{\text{ref}}$ (Lemma I.8), requiring Assumption H.3.

## H.6 Our Results

**Theorem H.5** (Formal Result of Theorem 6.1). *Fix $\delta \in (0,1)$, $m_{\text{off}} = K$, $m_{\text{on}} = 1$, suppose and the function class $\mathcal{C}$ follows Assumption H.1 w.r.t. $\pi^e$. Suppose the underlying MDP admits Bilinear rank $d$ on function class $\mathcal{C}$ and $d_{\text{ref}}$ on $\mathcal{C}_{\text{ref}}$, respectively, then with probability at least $1 - \delta$, Algorithm 2 obtains the following bound on cumulative suboptimality w.r.t. any comparator policy $\pi^e$:*

$$\sum_{t=1}^{K} V^{\pi^e} - V^{\pi^k} = \widetilde{O}\left(\min\left\{C_{\pi^e}^{\text{ref}} H\sqrt{dK\log\left(|\mathcal{F}|/\delta\right)},\ K\left(V^{\pi^e} - V^{\text{ref}}\right) + H\sqrt{d_{\text{ref}}K\log\left(|\mathcal{F}|/\delta\right)}\right\}\right).$$

(H.13)

Note that Theorem H.5 provides a guarantee for *any* comparator policy $\pi^e$, which can be directly applied to $\pi^\star$ described in our informal result (Theorem 6.1). We also change the notation for the reference policy from $\mu$ in Theorem 6.1 to $\pi^{\text{ref}}$ (similarly, $d_{\text{ref}}, V^{\text{ref}}, C_{\pi^e}^{\text{ref}}$ correspond to $d_\mu, V^\mu, C_{\pi^e}^\mu$ in Theorem 6.1) for notation consistency in the proof. Our proof of Theorem H.5 largely follows the proof of Theorem 1 of [52], and the major changes are caused by Assumption H.1, Assumption H.3, and Definition H.4.

*Proof.* Let $\mathcal{E}_k$ denote the event that $\{f_0^k(s,a) \leq Q^{\text{ref}}(s,a)\}$ and $\bar{\mathcal{E}}_k$ denote the event that $\{f_0^k(s,a) > Q^{\text{ref}}(s,a)\}$. Let $V^{\text{ref}}(s) = \max_a Q^{\text{ref}}(s,a)$, we start by noting that

$$\sum_{k=1}^{K} V^{\pi^e} - V^{\pi^{f^k}} = \sum_{k=1}^{K} \mathbb{E}_{s\sim\rho}\left[V_0^{\pi^e}(s) - V_0^{\pi^{f^k}}(s)\right]$$

$$= \underbrace{\sum_{k=1}^{K} \mathbb{E}_{s\sim\rho}\left[\mathbb{1}\left\{\bar{\mathcal{E}}_k\right\}\left(V_0^{\pi^e}(s) - V^{\text{ref}}(s)\right)\right]}_{\Gamma_0} + \underbrace{\sum_{k=1}^{K} \mathbb{E}_{s\sim\rho}\left[\mathbb{1}\left\{\bar{\mathcal{E}}_k\right\}\left(V^{\text{ref}}(s) - \max_a f_0^k(s,a)\right)\right]}_{=0,\text{ by the definition of }\bar{\mathcal{E}}_k\text{ and line 9 of Algorithm 2}}$$

$$+ \underbrace{\sum_{t=1}^{K} \mathbb{E}_{s\sim\rho}\left[\mathbb{1}\left\{\bar{\mathcal{E}}_k\right\}\left(\max_a f_0^k(s,a) - V_0^{\pi^{f^k}}(s)\right)\right]}_{\Gamma_1} + \underbrace{\sum_{k=1}^{K} \mathbb{E}_{s\sim\rho}\left[\mathbb{1}\left\{\mathcal{E}_k\right\}\left(V_0^{\pi^e}(s) - \max_a f_0^k(s,a)\right)\right]}_{\Gamma_2}$$

$$+ \underbrace{\sum_{t=1}^{T} \mathbb{E}_{s\sim\rho}\left[\mathbb{1}\left\{\mathcal{E}_k\right\}\left(\max_a f_0^k(s,a) - V_0^{\pi^{f^k}}(s)\right)\right]}_{\Gamma_3}.$$

(H.14)

Let $K_1 = \sum_{k=1}^{K} \mathbb{1}\left\{f_0^k(s,a) > Q^{\text{ref}}(s,a)\right\}$ and $K_2 = \sum_{k=1}^{K} \mathbb{1}\left\{f_0^k(s,a) \leq Q^{\text{ref}}(s,a)\right\}$ (or equivalently $K_1 = \sum_{k=1}^{K} \mathbb{1}\left\{\bar{\mathcal{E}}_k\right\}$, $K_2 = \sum_{k=1}^{K} \mathbb{1}\left\{\mathcal{E}_k\right\}$). For $\Gamma_0$, we have

$$\Gamma_0 = K_2 \mathbb{E}_{s\sim\rho}\left(V^{\pi^e}(s) - V^{\text{ref}}(s)\right).$$

(H.15)

For $\Gamma_2$, we have

$$\Gamma_2 = \sum_{k=1}^{K} \mathbb{E}_{s \sim \rho} \left[ \mathbb{1}\left\{\mathcal{E}_k\right\} \left( V_0^{\pi^e}(s) - \max_a f_0^k(s,a) \right) \right]$$

$$\overset{(i)}{\leq} \sum_{k=1}^{K} \mathbb{1}\left\{\mathcal{E}_k\right\} \sum_{h=0}^{H-1} \mathbb{E}_{s,a \sim d_h^{\pi^e}} \left[ \mathcal{T} f_{h+1}^k(s,a) - f_h^k(s,a) \right] \tag{H.16}$$

$$\overset{(ii)}{\leq} \sum_{k=1}^{K} \left[ C_{\pi^e}^{\mathrm{ref}} \cdot \mathbb{1}\left\{\mathcal{E}_k\right\} \sqrt{\sum_{h=0}^{H-1} \mathbb{E}_{s,a \sim \nu_h} \left[ \left( f_h^k(s,a) - \mathcal{T} f_{h+1}^k(s,a) \right)^2 \right]} \right]$$

$$\overset{(iii)}{\leq} K_1 C_{\pi^e}^{\mathrm{ref}} \sqrt{H \cdot \Delta_{\mathrm{off}}},$$

where $\Delta_{\mathrm{off}}$ is similarly defined as Song et al. [52] (See (I.3) of Lemma I.4). Inequality $(i)$ holds because of Lemma I.5, inequality $(ii)$ holds by the definition of $C_{\pi^e}^{\mathrm{ref}}$ (Definition H.4), inequality $(iii)$ holds by applying Lemma I.4 with the function class satisfying Assumption H.1, and Definition H.4. Note that the telescoping decomposition technique in the above equation also appears in [58, 24, 9]. Next, we will bound $\Gamma_1 + \Gamma_3$:

$$\Gamma_1 + \Gamma_3 = \sum_{k=1}^{K} \left( \mathbb{1}\left\{\mathcal{E}_k\right\} + \mathbb{1}\left\{\bar{\mathcal{E}}_k\right\} \right) \mathbb{E}_{s \sim d_0} \left[ \max_a f_0^k(s,a) - V_0^{\pi^{f^k}}(s) \right]$$

$$\overset{(i)}{\leq} \sum_{k=1}^{K} \left( \mathbb{1}\left\{\mathcal{E}_k\right\} + \mathbb{1}\left\{\bar{\mathcal{E}}_k\right\} \right) \sum_{h=0}^{H-1} \left| \mathbb{E}_{s,a \sim d_h^{\pi^{f^k}}} \left[ f_h^k(s,a) - \mathcal{T} f_{h+1}^k(s,a) \right] \right| \tag{H.17}$$

$$\overset{(ii)}{=} \sum_{t=1}^{K} \left[ \left( \mathbb{1}\left\{\mathcal{E}_k\right\} + \mathbb{1}\left\{\bar{\mathcal{E}}_k\right\} \right) \sum_{h=0}^{H-1} \left| \left\langle X_h(f^k), W_h(f^k) \right\rangle \right| \right]$$

$$\overset{(iii)}{\leq} \sum_{k=1}^{K} \left[ \left( \mathbb{1}\left\{\mathcal{E}_k\right\} + \mathbb{1}\left\{\bar{\mathcal{E}}_k\right\} \right) \sum_{h=0}^{H-1} \left\| X_h(f^k) \right\|_{\Sigma_{k-1;h}^{-1}} \sqrt{\Delta_{\mathrm{on}} + \lambda B_W^2} \right],$$

where $\Delta_{\mathrm{on}}$ is similarly defined as Song et al. [52] (See (I.4) of Lemma I.4). Inequality $(i)$ holds by Lemma I.6, equation $(ii)$ holds by the definition of Bilinear model ((H.8) in Definition H.2), inequality $(iii)$ holds by Lemma I.7 and Lemma I.4 with the function class satisfying Assumption H.1. Using Lemma I.8, we have that

$$\Gamma_1 + \Gamma_3$$

$$\leq \sum_{k=1}^{K} \left[ \left( \mathbb{1}\left\{\mathcal{E}_k\right\} + \mathbb{1}\left\{\bar{\mathcal{E}}_k\right\} \right) \sum_{h=0}^{H-1} \left\| X_h(f^k) \right\|_{\Sigma_{k-1;h}^{-1}} \sqrt{\Delta_{\mathrm{on}} + \lambda B_W^2} \right]$$

$$\overset{(i)}{\leq} H \sqrt{2d \log\left(1 + \frac{K_1 B_X^2}{\lambda d}\right) \cdot (\Delta_{\mathrm{on}} + \lambda B_W^2) \cdot K_1} + H \sqrt{2d_{\mathrm{ref}} \log\left(1 + \frac{K_2 B_X^2}{\lambda d_{\mathrm{ref}}}\right) \cdot (\Delta_{\mathrm{on}} + \lambda B_W^2) \cdot K_2}$$

$$\overset{(ii)}{\leq} H \left( \sqrt{2d \log\left(1 + \frac{K_1}{d}\right) \cdot (\Delta_{\mathrm{on}} + B_X^2 B_W^2) \cdot K_1} + \sqrt{2d_{\mathrm{ref}} \log\left(1 + \frac{K_2}{d_{\mathrm{ref}}}\right) \cdot (\Delta_{\mathrm{on}} + B_X^2 B_W^2) \cdot K_2} \right), \tag{H.18}$$

where the first part of inequality $(i)$ holds by the assumption that the underlying MDPs have bellman rank $d$ (Definition H.2) when $\bar{\mathcal{E}}_k$ happens, and the second part of inequality $(i)$ holds by the assumption that $\mathcal{C}_{\mathrm{ref}}$ has bilinear rank $d_{\mathrm{ref}}$ (Assumption H.3) $\mathcal{C}_{\mathrm{ref}}$ has bellman rank $d_{\mathrm{ref}}$ when $\mathcal{E}_k$ happens. Inequality $(ii)$ holds by plugging in $\lambda = B_X^2$. Substituting (H.15), inequality H.16, and inequality (H.18) into (H.14), we have

$$\sum_{t=1}^{K} V^{\pi^e} - V^{\pi^{f^k}} \leq \Gamma_0 + \Gamma_2 + \Gamma_1 + \Gamma_3 \leq K_2 \left( V^{\pi^e}(s) - V^{\mathrm{ref}}(s) \right) + K_1 C_{\pi^e}^{\mathrm{ref}} \sqrt{H \cdot \Delta_{\mathrm{off}}}$$

$$+ H \left( \sqrt{2d \log\left(1 + \frac{K_1}{d}\right) \cdot (\Delta_{\mathrm{on}} + B_X^2 B_W^2) \cdot K_1} + \sqrt{2d_{\mathrm{ref}} \log\left(1 + \frac{K_2}{d_{\mathrm{ref}}}\right) \cdot (\Delta_{\mathrm{on}} + B_X^2 B_W^2) \cdot K_2} \right) \tag{H.19}$$

Plugging in the values of $\Delta_{\text{on}}, \Delta_{\text{off}}$ from (I.3) and (I.4), and using the subadditivity of the square root function, we have

$$\sum_{k=1}^{K} V^{\pi^e} - V^{\pi^{f^k}}$$

$$\leq K_2 \left( V^{\pi^e}(s) - V^{\text{ref}}(s) \right) + 16 V_{\max} C_{\pi^e}^{\text{ref}} K_1 \sqrt{\frac{H}{m_{\text{off}}} \log \left( \frac{2HK_1 |\mathcal{F}|}{\delta} \right)}$$

$$+ \left( 16 V_{\max} \sqrt{\frac{1}{m_{\text{on}}} \log \left( \frac{2HK_1 |\mathcal{F}|}{\delta} \right)} + B_X B_W \right) \cdot H \sqrt{2dK_1 \log \left( 1 + \frac{K_1}{d} \right)} \quad \text{(H.20)}$$

$$+ \left( 16 V_{\max} \sqrt{\frac{1}{m_{\text{on}}} \log \left( \frac{2HK_2 |\mathcal{F}|}{\delta} \right)} + B_X B_W \right) \cdot H \sqrt{2d_{\text{ref}} K_2 \log \left( 1 + \frac{K_2}{d_{\text{ref}}} \right)}.$$

Setting $m_{\text{off}} = K, m_{\text{on}} = 1$ in the above equation completes the proof, we have

$$\sum_{k=1}^{K} V^{\pi^e} - V^{\pi^k}$$

$$\leq \widetilde{O} \left( C_{\pi^e}^{\text{ref}} \sqrt{HK_1 \log (|\mathcal{F}|/\delta)} \right) + \widetilde{O} \left( H \sqrt{dK_1 \log (|\mathcal{F}|/\delta)} \right)$$

$$+ K_2 \left( V^{\pi^e}(s) - V^{\text{ref}}(s) \right) + \widetilde{O} \left( H \sqrt{d_{\text{ref}} K_2 \log (|\mathcal{F}|/\delta)} \right) \quad \text{(H.21)}$$

$$\leq \begin{cases} \widetilde{O} \left( C_{\pi^e}^{\text{ref}} H \sqrt{dK_1 \log (|\mathcal{F}|/\delta)} \right) & \text{if } K_1 \gg K_2, \\ \widetilde{O} \left( K_2 \left( V^{\pi^e} - V^{\text{ref}} \right) + H \sqrt{d_{\text{ref}} K_2 \log (|\mathcal{F}|/\delta)} \right) & \text{otherwise.} \end{cases}$$

$$\leq \widetilde{O} \left( \min \left\{ C_{\pi^e}^{\text{ref}} H \sqrt{dK \log (|\mathcal{F}|/\delta)}, \ K \left( V^{\pi^e} - V^{\text{ref}} \right) + H \sqrt{d_{\text{ref}} K \log (|\mathcal{F}|/\delta)} \right\} \right),$$

where the last inequality holds because $K_1, K_2 \leq K$, which completes the proof. $\qquad\square$

# I   Key Results of HyQ [52]

In this section, we restate the major theoretical results of Hy-Q [52] for completeness.

## I.1   Assumptions

**Assumption I.1** (Realizability and Bellman completeness). *For any $h$, we have $Q_h^\star \in \mathcal{F}_h$, and additionally, for any $f_{h+1} \in \mathcal{F}_{h+1}$, we have $\mathcal{T} f_{h+1} \in \mathcal{F}_h$.*

**Definition I.2** (Bellman error transfer coefficient). *For any policy $\pi$, we define the transfer coefficient as*

$$C_\pi := \max \left\{ 0, \max_{f \in \mathcal{F}} \frac{\sum_{h=0}^{H-1} \mathbb{E}_{s,a \sim d_h^\pi}[\mathcal{T} f_{h+1}(s,a) - f_h(s,a)]}{\sqrt{\sum_{h=0}^{H-1} \mathbb{E}_{s,a \sim \nu_h}(\mathcal{T} f_{h+1}(s,a) - f_h(s,a))^2}} \right\}. \quad \text{(I.1)}$$

## I.2   Main Theorem of Hy-Q

**Theorem I.3** (Theorem 1 of Song et al. [52]). *Fix $\delta \in (0,1), m_{\text{off}} = K, m_{\text{on}} = 1$, and suppose that the underlying MDP admits Bilinear rank $d$ (Definition H.2), and the function class $\mathcal{F}$ satisfies Assumption I.1. Then with probability at least $1 - \delta$, HyQ obtains the following bound on cumulative suboptimality w.r.t. any comparator policy $\pi^e$:*

$$\text{Reg}(K) = \widetilde{O} \left( \max\{C_{\pi^e}, 1\} V_{\max} B_X B_W \sqrt{dH^2 K \cdot \log(|\mathcal{F}|/\delta)} \right). \quad \text{(I.2)}$$

## I.3   Key Lemmas

### I.3.1   Least Squares Generalization and Applications

**Lemma I.4** (Lemma 7 of Song et al. [52], Online and Offline Bellman Error Bound for FQI). *Let $\delta \in (0,1)$ and $\forall h \in [H-1], k \in [K]$, let $f_h^{k+1}$ be the estimated value function for time step $h$*

*computed via least square regression using samples in the dataset $\{\mathcal{D}_h^\nu, \mathcal{D}_h^1, \ldots, \mathcal{D}_h^T\}$ in (H.1) in the iteration $t$ of Algorithm 2. Then with probability at least $1 - \delta$, for any $h \in [H-1]$ and $k \in [K]$, we have*

$$\left\| f_h^{k+1} - \mathcal{T} f_{h+1}^{k+1} \right\|_{2,\nu_h}^2 \leq \frac{1}{m_{\text{off}}} 256 V_{\max}^2 \log(2HK|\mathcal{F}|/\delta) =: \Delta_{\text{off}} \qquad \text{(I.3)}$$

*and*

$$\sum_{\tau=1}^{k} \left\| f_h^{k+1} - \mathcal{T} f_{h+1}^{k+1} \right\|_{2,\mu_h^\tau}^2 \leq \frac{1}{m_{\text{on}}} 256 V_{\max}^2 \log(2HK|\mathcal{F}|/\delta) =: \Delta_{\text{on}}, \qquad \text{(I.4)}$$

*where $\nu_h$ denotes the offline data distribution at time $h$, and the distribution $\mu_h^\tau \in \Delta(s,a)$ is defined such that $s, a \sim d_h^{\pi^\tau}$.*

### I.3.2 Bounding Offline Suboptimality via Performance Difference Lemma

**Lemma I.5** (Lemma 5 of Song et al. [52], performance difference lemma of w.r.t. $\pi^e$)**.** *Let $\pi^e = (\pi_0^e, \ldots, \pi_{H-1}^e)$ be a comparator policy and consider any value function $f = (f_0, \ldots, f_{H-1})$, where $f_h : \mathcal{S} \times \mathcal{A} \mapsto \mathbb{R}$. Then we have*

$$\mathbb{E}_{s \sim d_0}\left[ V_0^{\pi^e}(s) - \max_a f_0(s,a) \right] \leq \sum_{i=1}^{H-1} \mathbb{E}_{s,a \sim d_i^{\pi^e}}\left[ \mathcal{T} f_{i+1}(s,a) - f_i(s,a) \right], \qquad \text{(I.5)}$$

*where we define $f_H(s,a) = 0, \forall(s,a)$.*

### I.3.3 Bounding Online Suboptimality via Performance Difference Lemma

**Lemma I.6** (Lemma 4 of Song et al. [52], performance difference lemma)**.** *For any function $f = (f_0, \ldots, f_{H-1})$ where $f_h : \mathcal{S} \times \mathcal{A} \mapsto \mathbb{R}$ and $h \in [H-1]$, we have*

$$\mathbb{E}_{s \sim d_0}\left[ \max_a f_0(s,a) - V_0^{\pi^f}(s) \right] \leq \sum_{h=0}^{H-1} \left| \mathbb{E}_{s,a \sim d_h^{\pi^f}}\left[ f_h(s,a) - \mathcal{T} f_{h+1}(s,a) \right] \right|, \qquad \text{(I.6)}$$

*where we define $f_H(s,a) = 0, \forall s, a$.*

**Lemma I.7** (Lemma 8 of Song et al. [52], upper bounding bilinear class)**.** *For any $k \geq 2$ and $h \in [H-1]$, we have*

$$\left| \langle W_h(f^k), X_h(f^k) \rangle \right| \leq \left\| X_h(f^k) \right\|_{\Sigma_{k-1;h}^{-1}} \sqrt{\sum_{i=1}^{k-1} \mathbb{E}_{s,a \sim d_h^{f^i}}\left[ \left( f_h^k - \mathcal{T} f_{h+1}^k \right)^2 \right] + \lambda B_W^2}, \qquad \text{(I.7)}$$

*where $\Sigma_{k-1;h}$ is defined as (H.5) and we use $d_h^{f^k}$ to denote $d_h^{\pi^{f^k}}$.*

**Lemma I.8** (Lemma 6 of Song et al. [52], bounding the inverse covariance norm)**.** *Let $X_h(f^1), \ldots, X_h(f^K) \in \mathbb{R}^d$ be a sequence of vectors with $\left\| X_h(f^k) \right\|_2 \leq B_X < \infty, \forall k \leq K$. Then we have*

$$\sum_{k=1}^{K} \left\| X_h(f^k) \right\|_{\Sigma_{k-1;h}^{-1}} \leq \sqrt{2dK \log\left( 1 + \frac{KB_X^2}{\lambda d} \right)}, \qquad \text{(I.8)}$$

*where we define $\Sigma_{k;h} := \sum_{\tau=1}^{k} X_h(f^\tau) X_h(f^\tau)^T + \lambda I$ and we assume $\lambda \geq B_X^2$ holds $\forall k \in [K]$.*

