# OpenReview forum: "Cal-QL: Calibrated Offline RL Pre-Training for Efficient Online Fine-Tuning"
_NeurIPS.cc/2023/Conference — NeurIPS 2023 poster_

### Official Review · Reviewer_J85q · 2023-07-06

**Soundness:** 3 good
**Presentation:** 2 fair
**Contribution:** 2 fair
**Rating:** 5
**Confidence:** 2

**Summary:**

The paper introduces a method called Calibrated Q-learning (Cal-QL) for efficient online fine-tuning. This method learns an effective initialization from offline data, which enables fast online fine-tuning. It does so by underestimating the value of the learned policy from offline data while ensuring that the learned Q-values are at a reasonable scale.

**Strengths:**

This work focuses on the offline pre-training phase for better online fine-tuning and identifies why CQL unlearn initially, which is new to me. The paper is well-written and easy to follow.

**Weaknesses:**

My main concerns are listed below:

1. Does the proposed approach increase the possibility of overestimation, given that it selects values larger than the behaviour policy? This question particularly arises when dealing with datasets that lack diversity or expert data.
2. The baselines, such as CQL and AWAC, were tested on locomotion tasks, such as Hopper, Walker2d, and HalfCheetah. However, I did not see experiment results on these datasets. Does your approach perform poorly on these tasks?
3. Some newly presented offline-to-online baselines, such as off2on[1], are not included.

[1] [Offline-to-Online Reinforcement Learning via Balanced Replay and Pessimistic Q-Ensemble

**Questions:**

Why not conduct experiments on continuous control tasks similar to those performed by CQL and AWAC?

---

> ### Author Rebuttal · Authors · 2023-08-10
>
> Thank you for your detailed feedback and for a positive assessment of our work. To address your concerns, we have now run experiments evaluating Cal-QL on Gym locomotion tasks and found that Cal-QL outperforms prior methods on these tasks, tied with CQL. We also added a comparison to SPOT (Wu et al. 2023), a state-of-the-art offline pre-training and online fine-tuning method similar to Off2On. Regarding Off2On (Lee et al. 2022), we note that the author’s published code only contains the online fine-tuning portion of the method and does not provide the code for pre-training. In the limited time for this rebuttal, we tried our best to implement the offline pre-training part but were not able to get it running. But nonetheless, we agree this is a valuable comparison and we will aim to get this method for the final version of the paper.
>
>
> We answer the rest of your questions in detail below. **Please let us know if your concerns are addressed, and if so, we would appreciate it if you are willing to upgrade your score. We are happy to discuss further and answer any remaining concerns.**
> ___
>
> > **The baselines, such as CQL and AWAC, were tested on locomotion tasks, such as Hopper, Walker2d, and HalfCheetah. However, I did not see experiment results on these datasets. Does your approach perform poorly on these tasks?**
>
> To address this concern, we have now added results on these tasks in **Figure B** in the one page PDF for Cal-QL, CQL, IQL, AWAC, and Hybrid RL. In general, we observe that Cal-QL attains the best performance on these tasks, tied with CQL (within the standard error margins). Our setup is as follows: following the scheme in prior work (Off2On, Lee et al. 2022) which finds non-conservative fine-tuning to perform best on top of the pre-trained CQL Q-function on the locomotion tasks (albeit it studies the deprecated -v0 versions of these tasks), we set the conservatism hyperparameter $\alpha$ to 0 for both CQL and Cal-QL during fine-tuning. For fair comparisons, we also tune the value of the conservatism hyperparameter for all other methods (IQL, AWAC) and pick the best value. Clearly, the trends in the result demonstrate the superior performance of Cal-QL over other state-of-the-art fine-tuning (IQL, AWAC) as well as fast online RL methods (Hybrid RL), that do not employ separate pre-training.
> ___
> > **Some newly presented offline-to-online baselines, such as off2on[1], are not included.**
>
> As discussed earlier, we have now added an experiment comparing Cal-QL to a new method, SPOT (Wu et al. 2022) on the antmaze domains and found that Cal-QL attains better final performance and cumulative regret. Regarding Off2On (Lee et al. 2022), the published code only contains the online fine-tuning part and does not provide the code for pre-training. In the limited time for this rebuttal, we tried our best to implement the offline pre-training part but were not able to get it running. Additionally, due to differences in the codebase infrastructure, we cannot directly utilize pre-trained checkpoints from our code directly in the Off2On codebase. We will aim to get this method for the final version of the paper.
> |   env      |    Cal-QL (Normalized score)   |   SPOT (Normalized score)    |   Cal-QL (Regret)    | SPOT (Regret) |
> |------------|:----------------------------------------:|:---------------------------------------:|:-------------------------:|:------------------:|
> | large-diverse | 33 - **95**   | 17.5 - 81.0  |  **0.20**  | 0.23  |
> | large-play | 26 - **90**   | 31.5 - 87.0  |  **0.28**  | 0.29  |
> | medium-diverse | 75 - **98**  |  73.7 - 94.5 | 0.05  | 0.05  |
> | medium-play |  54 - 97 | 67.2 - **97.2**  | 0.07  |  **0.05** |
> ___
> > **Does the proposed approach increase the possibility of overestimation, given that it selects values larger than the behavior policy? This question particularly arises when dealing with datasets that lack diversity or expert data.**
>
> This is a great question! We do observe that for a given value of the conservatism hyperparameter $\alpha$, the Q-values learned by Cal-QL are higher than those learned by base CQL, which tend to be quite heavily underestimated in datasets that lack diversity. This does not mean that the Q-values are necessarily overestimated with respect to their ground-truth value (i.e., the Q-function of  the _learned policy_), even in the setting with narrow data, because the learned policy can still improve upon the behavior policy in a narrow dataset (for example, consider the D4RL locomotion medium datasets).
> In the special case of expert data, such as on the Adroit tasks, we did observe that the Q-values learned by Cal-QL were larger than the ground-truth expert behavior policy return, but we believe this is not hurtful to performance: for attaining good offline performance, all we need is for the underlying method to create sufficient gap between Q-values at out-of-distribution and in-distribution actions. We always observe in our experiments that Cal-QL is able to induce a similar gap as CQL, thus preserving the relative ordering on actions. Moreover, a bit of overestimation may in fact be helpful during online fine-tuning, especially to keep the learned  policy close to the offline expert data.
> Indeed, we find that empirically, this over-estimation was not a problem: note that in all of our results, the performance of CQL and Cal-QL in the offline phase were quite similar to each other, and during fine-tuning, Cal-QL outperforms CQL by alleviating or reducing the performance dip. We have added this discussion in the paper now.

---

> ### Author Response · Authors · 2023-08-16
>
> Dear Reviewer J85q,
>
> Thank you for your feedback! We were wondering if you have gotten a chance to go over our responses -- we have now added the results on D4RL locomotion datasets, added an experiment to compare Cal-QL to SPOT (Wu et al 2022, a baseline similar to Off2On), and also provided clarifications on your other concerns. We would appreciate it if you could have a look. We are happy to discuss further.
>
> Thanks!

---

> > ### Author Response · Authors · 2023-08-20
> > **Any remaining concerns**
> >
> > Dear Reviewer J85q,
> >
> > As the author-reviewer discussion period draws to the close in less than 48 hours, we were hoping to check in with you if you have gotten a chance to go over our responses and if your concerns are addressed. We have performed several additional experiments (details in the rebuttal PDF) and would be grateful if you engage in a discussion with us. We are happy to address any remaining concerns. Thank you so much!

---

### Official Review · Reviewer_zhZZ · 2023-07-06

**Soundness:** 3 good
**Presentation:** 3 good
**Contribution:** 2 fair
**Rating:** 5
**Confidence:** 4

**Summary:**

The paper proposed a new approach to Offine-RL Online Fine-Tuning called calibrated Q-learning (Cal-QL). This approach aims to learn an effective initialization from offline data that enables fast online fine-tuning capabilities. Cal-QL accomplishes this by learning a conservative value function initialization that underestimates the value of the learned policy from offline data while ensuring that the learned Q-values are at a reasonable scale. The contributions of this paper include: 1) The paper theoretically analyzes the cumulative regret obtained by online fine-tuning, when the value function pre-training with Cal-QL. 2) Empirically, Cal-QL outperforms previous methods on 9/11 fine-tuning benchmark tasks.

**Strengths:**

1.	Writing is clear and understandable.
2.	The paper proposes a new definition (calibration) and gives the theoretical analysis of Cal-QL.
3.	The method can be implemented on top of conservative Q-learning (CQL) for offline RL with few code changes.
4.	The Cal-QL is better than the baseline methods in most tasks.


**Weaknesses:**

1.	The assumption of Cal-QL seems a bit too strong, that Cal-QL will be particularly effective in controlling the efficiency of online fine-tuning only when the reference policy μ is close to the narrow expert policy.
2.	Cal-QL is also most related to methods that utilize a pessimistic RL algorithm for offline training but incorporate exploration in fine-tuning [1,2]. However, these two important baseline methods are missing from the experimental comparison results (Figure 6, Table 1 and 2).
3.	The coefficient (\alpha) of the conservative term (in equation 3.1) is an important parameter that directly affects the experimental results. However, the hyperparameter (\alpha) ablation experiments are missing.


**Questions:**

1.	How many random seeds are used for the experimental results of Table 1 and Table 2? Same as Figure 6, six random seeds? Moreover, how many random seeds are used in Figure 7 and 8?
2.	In Appendix Table 3, for the visual-manipulation task, the paper utilized separate values for offline (\alpha = 5) and online (\alpha = 0.5) phases for the final results. Is the proposed algorithm sensitive to hyperparameters (\alpha and mixing ratio)? To what extent do the final experimental results of different tasks depend on the hyperparameters?


**Limitations:**

1.	It would be better if there are more comparative experiments with important baseline methods [1,2].
2.	To test the sensitivity of the method to hyperparameters, there should be more experimental comparisons and analyses of hyperparameters \alpha and mixing ratio.

References:

[1] S. Lee, Y. Seo, K. Lee, P. Abbeel, and J. Shin. Offline-to-online reinforcement learning via balanced replay and pessimistic q-ensemble. In Conference on Robot Learning, pages 1702–1712. PMLR, 2022.

[2] J. Wu, H. Wu, Z. Qiu, J. Wang, and M. Long. Supported policy optimization for offline reinforcement learning. arXiv preprint arXiv:2202.06239, 2022.

---

> ### Author Rebuttal · Authors · 2023-08-10
>
> Thank you for your review and for a positive assessment of our work. To address your concern regarding comparisons, we have added an experiment to compare Cal-QL to SPOT (Wu et al 2022), as you suggested. We found that Cal-QL outperforms this prior method (more details below). We also clarify that we already compare to O3F which performs pessimistic offline RL training followed by fine-tuning with an exploration bonus in **Figure 6** and **Tables 1 & 2** in the submission and find that Cal-QL outperforms this method as well. In Figure D, we also add an ablation study over $\alpha$ for Cal-QL and find that Cal-QL is no more sensitive to $\alpha$ than CQL or other methods such as TD3+BC. We address the remaining concerns below.
>
>
> **Please let us know if your concerns are addressed and if so, we would be grateful if you could update your score. We would be happy to discuss further if any concerns remain.**
> ___
>
> > **Additional comparisons to methods that use pessimistic offline RL + optimism during online fine-tuning.**
>
> We have added a comparison to SPOT (Wu et al. 2022). We utilized the implementation of SPOT in the Clean Offline Reinforcement Learning (CORL) library (https://github.com/tinkoff-ai/CORL) and evaluated it on the 4 antmaze tasks. As shown below, we find that Cal-QL outperforms this approach, especially on the harder antmaze-large domains.
> |   env      |    Cal-QL (Normalized score)   |   SPOT (Normalized score)    |   Cal-QL (Regret)    | SPOT (Regret) |
> |------------|:--------------:|:---------------------------------------:|:-------------------------:|:------------------:|
> | large-diverse |33 - **95**|17.5 - 81.0| **0.20**  |0.23|
> |large-play|26 - **90**|31.5 - 87.0|**0.28**|0.29|
> | medium-diverse| 75 - **98**|  73.7 - 94.5 | 0.05  | 0.05  |
> | medium-play |54 - 97|67.2 - **97.2**  | 0.07  |  **0.05** |
>
> Regarding Off2On (Lee et al. 2022), the published code only contains code for online fine-tuning and does not provide the code for pre-training. In the limited time for this rebuttal, we tried our best to implement the offline pre-training part but were not able to get it running. Additionally, due to differences in the codebase infrastructure, we cannot directly utilize pre-trained checkpoints from our code directly in the Off2On codebase. Nonetheless, we agree this is a valuable baseline and will aim to get this method for the final version of the paper.
>
> Finally, also note that in Tables 1 & 2, Figure 6, we already compare to O3F (Mark et al. 2022), another approach that employs optimism during online training in conjunction with pessimistic offline RL pre-training and find that Cal-QL outperforms this approach as well.
> ___
>
> > **Ablation study for $\alpha$, online & offline alpha**
>
> This is an important point and we will add extensive ablation studies over these parameters for all methods in the final. During the rebuttal period, we ran an ablation on the Adroit door-binary task and found that setting $\alpha$ to its offline value was generally pretty close to the best-tuned $\alpha$ value, though other values of $\alpha$ can also work during the online fine-tuning phase. Empirically, we recommend that a practitioner can perform a sweep in the neighborhood of the offline alpha value to tune their method efficiently.
>
> Specifically, on the visual manipulation task, we present an ablation over alpha values for Cal-QL, CQL and TD3+BC in **Figure D** in the PDF. We observed that Cal-QL required the modification of the value of $\alpha$ to attain best results. However, this isn’t any different from prior methods such as CQL or TD3+BC, which also required this. IQL’s performance was relatively more robust on the expectile parameter $\beta$, but all of these $\beta$ values performed worse than half of the $\alpha$ values that we ran for Cal-QL. Thus, **Cal-QL is no more sensitive to $\alpha$ values than other state-of-the-art algorithms on this task.**
> ____
>
> > **Ablation study over mixing ratio.**
>
> We ran an experiment to evaluate the performance of both IQL and Cal-QL with different mixing ratios on the Adroit door-binary task. The results are shown in **Figure C** in the PDF. Perhaps as expected, we find an intuitive trend that for both methods, a smaller mixing ratio (i.e., training more on online data than offline data) improves faster than larger mixing ratios which entail more updates on offline data. **In summary, note that this trend in Cal-QL’s performance as a function of mixing ratio is no different than other state-of-the-art methods, such as IQL.** We will run a more extensive ablation study for the final version of the paper.
> ___
>
> > **Random seeds**
>
> We utilize 6 seeds for our results in Table 1 and 2 like in Figure 6. For Figures 7 & 8, we use 3 seeds. The high UTD experiments in Figure 7 are time-consuming, hence we only ran 3 seeds for these experiments.
> ___
>
> >**The assumption of Cal-QL seems a bit too strong, that Cal-QL will be particularly effective in controlling the efficiency of online fine-tuning only when the reference policy μ is close to the narrow expert policy.**
>
> We clarify that this is not an assumption, but rather a conclusion of our analysis for the special case when the reference policy is close to an expert policy. That is, we specifically stress that this is **not the __only__ case** where Cal-QL will be particularly effective, but rather one of the special cases that we chose to discuss in the paper. Theoretically, Cal-QL is effective whenever the bound on $\text{Reg}(K)$ in Theorem 6.1 is smaller than the guarantee of Song et al. 2023, even in other cases because we show that $C^\mu_{\pi^*}$ is smaller than the concentrability coefficient in Song et al. 2023.
> To corroborate the efficacy of Cal-QL with non-expert reference (behavior) policies, in our experiments in **Figure 6**, we demonstrate that Cal-QL is effective on antmaze-large and kitchen tasks, where the reference (behavior) policy is not expert.

---

> ### Author Response · Authors · 2023-08-16
>
> Dear Reviewer zhZZ,
>
> Thank you for your feedback! We were wondering if you have gotten a chance to go over our responses -- we have now added an experiment to compare Cal-QL to SPOT (Wu et al 2022), as you suggested., and also provided clarifications on your other concerns. We would appreciate it if you could have a look. We are happy to discuss further.
>
> Thanks!

---

> > ### Author Response · Authors · 2023-08-20
> > **Any remaining concerns**
> >
> > Dear Reviewer zhZZ,
> >
> > As the author-reviewer discussion period draws to the close in less than 48 hours, we were hoping to check in with you if you have gotten a chance to go over our responses and if your concerns are addressed. We have performed several additional experiments (details in the rebuttal PDF) and would be grateful if you engage in a discussion with us. We are happy to address any remaining concerns. Thank you so much!

---

### Official Review · Reviewer_zucz · 2023-07-07

**Soundness:** 3 good
**Presentation:** 3 good
**Contribution:** 3 good
**Rating:** 6
**Confidence:** 3

**Summary:**

The paper proposes a new offline to online algorithm, Cal-QL, that aims to mitigate the unlearning during the beginning of the online fine-tune stage. The paper first conducts empirical experiments to observe that the reason for the unlearning is due to overly pessimistic function values, and thus proposes the Cal-QL algorithm where lower bound the estimated value by the value of the reference policy. With the change to CQL, the proposed algorithm can outperform other Off2On algorithms on various tasks and shows consistent improvements, without the unlearning phenomena. The paper also proposes a theoretical algorithm and shows the regret bound under the bilinear model setting.

**Strengths:**

1. The paper investigates empirically the reason for the initial unlearning phenomena at the beginning of the fine-tuning stage, shows good evidence for the issue of overly pessimistic value estimation, and thus well motivate the calibrated algorithm design.

2. The paper compares with a good range of baselines in offline2online methods, and shows consistent improvements over the previous baselines, and indeed resolves the unlearning issue at the beginning of the fine-tuning stage, and thus provides good practical contribution.

3. The overall algorithm is simple and a good amount of implementation details are provided in the paper.

4. Some theory is also provided to make the paper more complete.

**Weaknesses:**

1. Although it is good to provide many baselines to compare, the current baselines presentation is slightly confusing. I understand that RLPD and HyQ are reasonable baselines to compare with, but the current paper is not clearly comparing with RLPD or HyQ? One would guess that Hybrid RL means HyQ and SAC+offline means RLPD, but it is not clear that in the experiments the paper is indeed comparing with the exact algorithms.

2. The theory algorithm seems to have some gap with the actual Cal-QL algorithm.

3. The completeness assumption in the calibrated case seems strong, because the paper is focusing on a certain strange subset of function class. For example, can one show the completeness assumption holds even in the tabular case?

**Questions:**

See above.

---

> ### Author Rebuttal · Authors · 2023-08-10
>
> Thank you for your feedback and a positive assessment of our work. We address your concerns below and will update the paper to clarify each of the questions.
>
> **Please let us know if your questions are resolved, and if so, we would be grateful if you are willing to upgrade your score. We are happy to discuss further if any questions are remaining.**
>
> ___
>
> > **the current baselines presentation is slightly confusing; the current paper is not clearly comparing with RLPD or HyQ.**
>
> To clarify, we do compare Cal-QL to the exact approaches of RLPD and HyQ in our experiments. The Hybrid RL approach in Figure 6 corresponds exactly to HyQ and we will update the legend of Figure 6 to clearly indicate this.
>
> Since RLPD has been specifically designed for high UTD RL, we compare to RLPD in all of our high UTD experiments in Figure 7. In our low UTD experiments (Figure 6), for fair comparison, we compare to ``SAC + offline data’’, a reduction of RLPD for the low UTD setting, which retains the 50:50 sampling between offline and online data, but does not utilize Q-function ensembles with layer normalization which is specifically required for addressing overestimation with high UTD. We will clarify these points in the updated version of the paper.
>
> ___
>
> > **theory algorithm seems to have some gap with the actual Cal-QL algorithm.**
>
> We will update the paper to discuss the limitation that our theoretical algorithm is a simplification of the practical algorithm. That said, we clarify that the goal of our theoretical analysis is to provide some intuition and motivation for our practical algorithm, rather than to provide a rigorous guarantee for our practical method. That said, we do note that this sort of a gap between theoretical algorithms and the practical algorithm is quite common in prior works: as one example, a closely related work Hybrid RL [3] (Song et al. ICLR 2023) also analyzes a version of FQI which minimizes TD-error exactly using an oracle and assumes access to all state-action pairs in theory, but the practical algorithm utilizes the DQN algorithm, which does not satisfy any of these conditions.
>
> Another example is ATAC (ICML 2022) [2], where the authors provide a theoretical version (Alg. 1) and a practical version (Alg. 2) of their algorithm. The regret analysis corresponds to the theoretical algorithm assuming the existence of a no-regret policy optimization oracle, which is hard to practically achieve. As for the practical version (Algorithm 2), ATAC also adopts an actor-critic type algorithm (with a pessimistic regularizer on the critic network) for updating the Q network and policy network. Definitely, addressing the gap between theory and practice is important for the RL community, but we believe that our contribution is akin to other works that develop practical algorithms supported by theoretical results for motivation.
>
> ___
>
> > **The completeness assumption in the calibrated case seems strong; For example, can one show the completeness assumption holds even in the tabular case?**
>
> Thank you for your question! Yes, in the tabular case, the completeness assumption B.1 holds if we consider the subset of the function class where the values in the learned table are bounded by the Q-value of the reference policy. For instance, if the reference policy is the optimal policy, then in the tabular setting a pessimistic function class can still represent all possible value functions.
>
> That said, we do agree with you that this sort of an assumption can be strong. Our rationale for this assumption was that it greatly simplified the exposition of our main insight (i.e., calibration during offline training can reduce cumulative regret incurred in online fine-tuning), without needing to handle the complexities with analyzing conservative value function training objectives like prior work (e.g., ATAC). We have already provided Appendix B.4 to elaborate on this connection and note that our analysis can be extended using the tools for handling conservatism in ATAC, which analyzes a pessimistic offline RL algorithm which utilizes a conservatism regularizer similar to CQL and Cal-QL, to avoid the need for this assumption. We have now noted this in the paper.
>
> [1] Xie, Tengyang, et al. "Bellman-consistent pessimism for offline reinforcement learning." Advances in neural information processing systems 34 (2021): 6683-6694.
> [2] Cheng, Ching-An, et al. "Adversarially trained actor critic for offline reinforcement learning." International Conference on Machine Learning. PMLR, 2022.
> [3] Song, Yuda, et al. "Hybrid RL: Using both offline and online data can make RL efficient." ICLR2023.

---

> ### Author Response · Authors · 2023-08-16
>
> Dear Reviewer zucz,
>
> Thank you for your feedback! We were wondering if you have gotten a chance to go over our responses -- we have now provided clarifications on your concerns. We would appreciate it if you could have a look. We are happy to discuss further.
>
> Thanks!

---

### Official Review · Reviewer_vsVq · 2023-07-08

**Soundness:** 3 good
**Presentation:** 4 excellent
**Contribution:** 3 good
**Rating:** 6
**Confidence:** 3

**Summary:**

Initially, this paper conducts an empirical investigation to understand why the offline policy derived from prior methods is inadequate as an initialization for online fine-tuning. Building upon these empirical findings and the derived hypothesis, the authors design Cal-QL, a method differing from CQL by only a single line, aiming to calibrate the Q function. The superiority of Cal-QL is demonstrated through both theoretical and empirical studies.

**Strengths:**

- The authors present meticulous empirical and theoretical explorations justifying the proposed calibration.
- The method, while simple, proves to be quite effective.
- The paper boasts commendable writing and organization.

**Weaknesses:**

- In section 4.1, Figure 3 shows a clear increase in estimated Q values correlating with the increase in normalized scores. However, it does not clarify whether this increase stems from in-distribution data, out-distribution data, or both. Therefore, the connection between the results in Figure 3 and the conclusion in lines 162-164, which suggest that the rise in estimated Q values is due to the in-distribution data, may not be logically stringent.
- Although the locomotion datasets in D4RL are extensively used as benchmarks, the authors did not test Cal-QL on these datasets. It would enhance the study if the authors could include comparisons with baselines on D4RL locomotion datasets.

**Questions:**

Please see Weaknesses.

**Limitations:**

The authors have adequately addressed the limitations and the potential negative societal impact of their work.

---

> ### Author Rebuttal · Authors · 2023-08-10
>
> Thank you for your review and positive feedback on the paper. To address your concerns, we have now added comparisons on D4RL locomotion tasks, where we find that Cal-QL outperforms prior methods IQL, AWAC, Hybrid RL, performing similarly to the best method, CQL. We have also clarified other concerns.
>
> **Please let us know if your concerns are addressed and if so, we would appreciate it if you could upgrade your score. We are happy to discuss further.**
>
> ___
>
> ## Results on D4RL locomotion datasets.
>
> We have now added results on these tasks in **Figure B** in the one page PDF for Cal-QL, CQL, IQL, AWAC, and Hybrid RL. In general, we observe that Cal-QL attains the best performance on these tasks, tied with CQL (within the standard error margins). Our setup is as follows: following the scheme in prior work (Off2On, Lee et al. 2022) which finds non-conservative fine-tuning to perform best on top of the pre-trained CQL Q-function on the locomotion tasks (albeit it studies the deprecated -v0 versions of the locomotion tasks), we set the conservatism hyperparameter $\alpha$ to 0 for both CQL and Cal-QL during fine-tuning. For fair comparisons, we also tune the value of the conservatism hyperparameter $\beta$ for all other methods (IQL, AWAC). Clearly, the trends in the result demonstrate the superior performance of Cal-QL over other state-of-the-art fine-tuning (IQL, AWAC) and fast online RL methods (Hybrid RL).
>
> ## Regarding Figure 3.
>
> We apologize for the confusion. We have now plotted the average expected Q-value on states from the offline dataset (i.e., the in-distribution data) in **Figure A** in the PDF (we plot the placeholder value of 0 for Q-values on online data before fine-tuning begins at 50k steps). The Q-values on in-distribution data are labeled as `training/q_pi_offline_avg` and the Q-values on online data (i.e., out-of-distribution data) is labeled as `training/q_pi_online_avg`.
>
> Observe that the change in average Q-value plotted in Figure 3 of the submission stems from an increase in Q-values on both the in-distribution offline data and out-of-distribution data. Note crucially that the amount of increase on in-distribution Q-values is larger than the Q-values of online data. Therefore, the conclusion is that the CQL algorithm requires a few online samples to correct the heavily under-estimated in-distribution Q-values once it experiences online data logically follows from our experiment.

---

> > ### Comment · Reviewer_vsVq · 2023-08-19
> > **Rebuttal response**
> >
> > I am satisfied with the authors' comprehensive response, as it has resolved all of my issues. I recommend that they incorporate these new results and explanations into the updated version of the paper and include error bars in the newly introduced Figure B.
> >
> > With consideration of the thorough response, I'm pleased to increase my rating to 6.

---

> > > ### Author Response · Authors · 2023-08-20
> > > **Thank you for raising your score!**
> > >
> > > Thank you so much for replying to us and for raising your score! We are glad that your concerns are resolved -- we will incorporate these results and explanations into the final version of the paper, and also include error bars in Figure B. Thanks so much!

---

> ### Author Response · Authors · 2023-08-16
>
> Dear Reviewer vsVq,
>
> Thank you for your feedback! We were wondering if you have gotten a chance to go over our responses -- we have now added the results on D4RL locomotion datasets, and also provided clarifications on your other concerns. We would appreciate it if you could have a look. We are happy to discuss further.
>
> Thanks!

---

### Author Rebuttal · Authors · 2023-08-10

Dear Reviewers and the AC,

We thank all the reviewers for their detailed and constructive feedback. We are glad that Reviewer vsVq found our explanations to be meticulous, Reviewer zucz found our algorithm to be simple, and all reviewers found our paper to be well-written and clear.

To address the reviewers' concerns, we have added several new experiments in the rebuttal PDF attached to this response:

- In **Figure B**, we have added results comparing Cal-QL with other state-of-the-art fine-tuning methods (IQL, AWAC) and Hybrid RL on D4RL locomotion datasets and observe that outperforms these prior methods.

- In **Figures C and D**, we ran ablation experiments over $\alpha$ and mixing ratio to determine the sensitivity of Cal-QL to these parameters. We observed that Cal-QL is no more sensitive to the values of these parameters than existing state-of-the-art fine-tuning algorithms.

- In **Figure A**, we provided an additional plot to better understanding the unlearning phenomenon in CQL, supplementing Figure 3 in the main paper.

- In our response to **Reviewers zhZZ & J85q**, we also provide comparisons between Cal-QL and SPOT (Wu et al. 2022), a recently proposed online fine-tuning method on the antmaze tasks from D4RL. We observe that Cal-QL outperforms this prior method on the harder, antmaze-large tasks.

Additionally, we also provide several clarifications to address the reviewers' questions in our individual responses to each reviewer.

We believe that addressing the reviewers' feedback has improved the quality of the paper. We would be more than happy if the reviewers can let us know if their questions are answered. We will be happy to discuss further if there are any remaining questions.

---

### Decision · Program_Chairs · 2023-09-21

**Decision:**

Accept (poster)

**Comment:**

This paper is concerned with addressing the issue of unlearning during the online fine-tuning stage in offline to online reinforcement learning. The authors propose Cal-QL, a new algorithm that mitigates unlearning by lower bounding the estimated value using the reference policy, providing superior performance compared to other offline-to-online algorithms in empirical and theoretical studies. The idea looks interesting and novel, and the proposed method is sound with theoretical and experimental support.